# Differential trafficking of ligands trogocytosed via CD28 versus CTLA4 promotes collective cellular control of co-stimulation

Simon Zenke [1,2,9], Mauricio P. Sica[3], Florian Steinberg[4], Julia Braun[1,2], Alicia Zink[1], Alina Gavrilov[5,10], Alexander Hilger[6], Aditya Arra[7], Monika Brunner-Weinzierl [7], Roland Elling [6], Niklas Beyersdorf[8], Tim Lämmermann[5], Cristian R. Smulski[3] & Jan C. Rohr [1,11] ✉

Intercellular communication is crucial for collective regulation of cellular behaviors. While clustering T cells have been shown to mutually control the production of key communication signals, it is unclear whether they also jointly regulate their availability and degradation. Here we use newly developed reporter systems, bioinformatic analyses, protein structure modeling and genetic perturbations to assess this. We find that T cells utilize trogocytosis by competing antagonistic receptors to differentially control the abundance of immunoregulatory ligands. Specifically, ligands trogocytosed via CD28 are shuttled to the T cell surface, enabling them to co-stimulate neighboring T cells. In contrast, CTLA4-mediated trogocytosis targets ligands for degradation. Mechanistically, this fate separation is controlled by different acid-sensitivities of receptor-ligand interactions and by the receptor intracellular domains. The ability of CD28 and CTLA4 to confer different fates to trogocytosed ligands reveals an additional layer of collective regulation of cellular behaviors and promotes the robustness of population dynamics.

Upon infection T cells proliferate, differentiate, and ultimately either die or persist as memory cells, thereby forming a typical dynamic pattern of immune responses. For a given infection this pattern is remarkably reproducible, attesting to the robustness of immune responses. Conceptually, an immune response constitutes the sum of behaviors of cells contributing to an organism's reaction to a pathogenic microorganism. While this proposition sounds simple, the underlying principle is actually quite complex, because the behavior of immune cells is neither uniform, nor static, nor is it solely a cell-intrinsic property. Instead, it is highly heterogenous, can change rapidly and is shaped by communication among cells. Furthermore, there is no single master regulator orchestrating immune responses. Thus, immune responses emerge from a self-organizing complex adaptive network of dynamic interactions between cells exchanging information and other resources. Based on these considerations it becomes clear, that understanding how immune responses are formed

[1]Institute of Immunodeficiency, Medical Center and Faculty of Medicine, Albert-Ludwigs-University, Freiburg, Germany. [2]Faculty of Biology, Albert-Ludwigs-University, Freiburg, Germany. [3]Medical Physics Department, Centro Atómico Bariloche, Comisión Nacional de Energía Atómica (CNEA), Consejo Nacional de Investigaciones Científicas y Técnicas (CONICET), San Carlos de Bariloche, Argentina. [4]Center for Biological Systems Analysis, Albert-Ludwigs-University Freiburg, Freiburg, Germany. [5]Immune Cell Dynamics Group, Max Planck Institute of Immunobiology and Epigenetics, Freiburg, Germany. [6]Department of Pediatrics and Adolescent Medicine, Medical Center and Faculty of Medicine, Albert-Ludwigs-University, Freiburg, Germany. [7]Department of Experimental Pediatrics and Neonatology and Health Campus Immunology, Infectiology and Inflammation, Otto-von-Guericke-University, Magdeburg, Germany. [8]Institute of Virology and Immunobiology, University of Würzburg, Würzburg, Germany. [9]Present address: Matterhorn Biosciences GmbH, Basel, Switzerland. [10]Present address: Roche Pharmaceutical Research and Early Development, Roche Innovation Center Basel, Basel, Switzerland. [11]Present address: Novartis Institutes for Biomedical Research (NIBR), Novartis Pharma AG, Basel, Switzerland. ✉e-mail: jan.rohr@uniklinik-freiburg.de

will require the delineation of core concepts underlying such cellular collectivity.

We and others have documented that upon infection individual T lymphocytes expand and differentiate in a remarkably heterogeneous manner[1–3], but also employ mechanisms allowing for collective behavioral regulation[4,5]. This involves clustering of T lymphocytes, as well as directing the secretion of crucial immunoregulatory cytokines towards each other and tailoring it to local T cell population density[5–7]. However, cellular cooperation may not be restricted to jointly controlling the production of communication signals. Conceptually, the cellular collectivity landscape would be both more robust and more flexible if it also incorporated joint regulation of protein availability and degradation. This could be accomplished by trogocytosis, which denotes a process of receptor-mediated acquisition of transmembrane protein ligands expressed on neighboring cells[8].

Mechanistically, trogocytosis could involve blebbing of donor cell membrane fragments upon receptor-mediated endocytosis. This would lead to formation of donor cell-derived intraluminal vesicles (ILV) within endosomes of the recipient cell[9]. Alternatively, the molecule transfer could involve transient fusion of the surface membranes of adjacent cells[10], allowing the diffusion of transmembrane molecules from one cell to another. Such membrane fusion and endocytosis scenarios are not mutually exclusive, as it is entirely conceivable that fusion of cell surface membranes may precede receptor-mediated endocytosis or that within endosomes the membranes of intraluminal vesicles and those limiting the endosome might fuse[11]. Once transferred, acquired proteins are subject to the recipient cell's mechanisms of controlling protein turnover—and may be re-used or degraded. However, neither the mechanisms underlying the protein transfer, nor those regulating the fate of acquired proteins or their functional impact are well understood. Viewing trogocytosis from a cellular collectivity perspective, we set out to investigate these important aspects.

While immune cells are able to trogocytose a range of different proteins, we focused on CD80 and CD86 as these molecules are shared ligands for CD28 and CTLA4, which provide crucial co-stimulatory and inhibitory signals for T cells, respectively. Furthermore, we have previously demonstrated that CD80 and CD86 mediate collective behavioral regulation of clustered T cells[5]. Here we investigated whether activated CD8+ T cells employ trogocytosis to collectively regulate the availability and turnover of these ligands. For this, we developed several reporter systems and demonstrated that CD8+ T cells efficiently trogocytose CD80 and CD86 from antigen-presenting cells via CD28 and CTLA4 in a process involving little membrane fusion, but endocytosis. Furthermore, we showed that CD28 and CTLA4 confer different fates to trogocytosed ligands and that both the receptor intra- and extracellular domains play important roles in determining this fate-divergence. Finally, we documented that CD28-, but not CTLA4-mediated trogocytosis makes acquired molecules available at the cell surface, thereby enabling co-stimulation of neighboring cells. Together, our data establish trogocytosis-mediated control of protein availability as a mechanism for collective behavioral regulation of T cells and delineate critical parameters underlying this process.

## Results
### T cells acquire CD80 and CD86 from neighboring cells
We have previously shown how T cell expressed CD80 and CD86 regulates population dynamics within cell clusters forming around antigen-presenting cells (APCs)[5]. When we analyzed such clustering T cells, we noted that fluorescent antibodies directed against CD80 or CD86 stained Cd80−/−Cd86−/− T cells co-cultured with wild-type but not with Cd80−/−Cd86−/− APCs (Fig. 1a). As Cd80−/−Cd86−/− T cells cannot synthesize CD80 or CD86 themselves, this suggests that they had acquired these molecules from APCs and presented them on their cell surface.

We then investigated whether such trogocytosis of CD80 and CD86 by CD8+ T cells was merely an in vitro phenomenon or whether it also occurred upon infection in vivo. To this end we adoptively co-transferred P14 T cell receptor (TCR)-transgenic CD8+ T cells proficient or deficient for CD80 and CD86 into C57BL/6 mice. Subsequently, mice were infected with Lymphocytic Choriomeningitis virus (LCMV), whose gp33-epitope is recognized by the P14 TCR. We detected substantial amounts of CD80 and CD86 on transferred Cd80−/−Cd86−/− P14 T cells (Fig. 1b, Supplementary Fig. 1), demonstrating that also in vivo CD8+ T cells acquire CD80 and CD86 and display them on their surface. Consistent with some endogenous synthesis of CD80 and CD86 by wild-type T cells, their expression levels of both molecules were higher compared to Cd80−/−Cd86−/− T cells. Trogocytosis of CD80 and CD86 was only observed in T cells harvested from lymph nodes and spleen, but not from lung, liver or blood. This indicates that only in lymphatic organs T cells encounter sufficient numbers of CD80/CD86-expressing cells to acquire these ligands.

To further investigate this process, we generated artificial APCs expressing GFP-tagged CD80 molecules. Upon imaging CD8+ T cells clustering around these APCs, we noted that the GFP-signal was not confined to APCs, but was also detectable at the borders between and within T cells (Fig. 1c, Supplementary Movie 1). This corroborated that T cells had acquired CD80 from APCs and refuted that trogocytosis constituted an experimental artifact of separating the cells. Kinetic analyses showed that trogocytosis of CD80 and CD86 occurred quite rapidly (Fig. 1d). Concomitant TCR-triggering increased the amount of CD80/CD86-transfer, but was not required for this process (Fig. 1e). This observation fits well to previous publications describing that TCR-triggering can induce trogocytosis of non-recognized molecules located in the vicinity of pMHCs recognized[12–14]. Upon such bystander trogocytosis, the fate of acquired molecules is not determined by specific interactions with their receptors. Based on this, we performed subsequent experiments without concomitant TCR-triggering. While this setup may be perceived as simplistic, by limiting confounding effects of bystander trogocytosis it allows for a clearer delineation of the mechanisms underlying trogocytosis of CD80/CD86. In this setup trogocytosis of CD80/CD86 required specific receptor-ligand interaction, as mutations in the CD80/CD86 extracellular domains impairing receptor-interaction, as well as addition of the soluble CD80/CD86-ligand CTLA4-Fc abrogated their transfer (Fig. 1e, f).

Together our results demonstrate that clustering CD8+ T cells acquire essential co-stimulatory molecules from APCs and present them on their surface. Furthermore, they raise questions on the mechanisms underlying this process as well as on its biological function.

### Different fates of CD80 upon trogocytosis via CD28 and CTLA4
CD80 and CD86 are shared ligands for the receptors CD28 and CTLA4, both of which can be expressed by activated T cells. While several publications have reported that CD28 and CTLA4 are able to mediate trogocytosis of CD80 and CD86[15–17], at least two publications have denied CD28 this ability[17,18]. Consistent with naive CD8+ T cells expressing CD28, but not CTLA4, we observed less trogocytosis of CD80 and CD86 in naive Cd28−/−, but not Ctla4−/− CD8+ T cells (Fig. 2a, b). Likewise, antibody-mediated blockade of CD28, but not CTLA4 inhibited trogocytosis of CD80/CD86 (Fig. 2c, d). To directly compare the ability of CD28 and CTLA4 to mediate trogocytosis, we transduced T cells with either receptor and found both to increase the acquisition of CD80 (Fig. 2e). Transduction of B lymphocytes with CD28 or CTLA4 sufficed to enable them to trogocytose CD80 (Supplementary Fig. 2a). Conversely, T cell lines expressing neither CD28 nor CTLA4 did not trogocytose CD80 and CD86, and this was restored by expressing either CD28 or CTLA4 (Supplementary Fig. 2b, c). Together these experiments demonstrate that expression of CD28 or CTLA4 is necessary and sufficient for trogocytosis of CD80 and CD86.

 

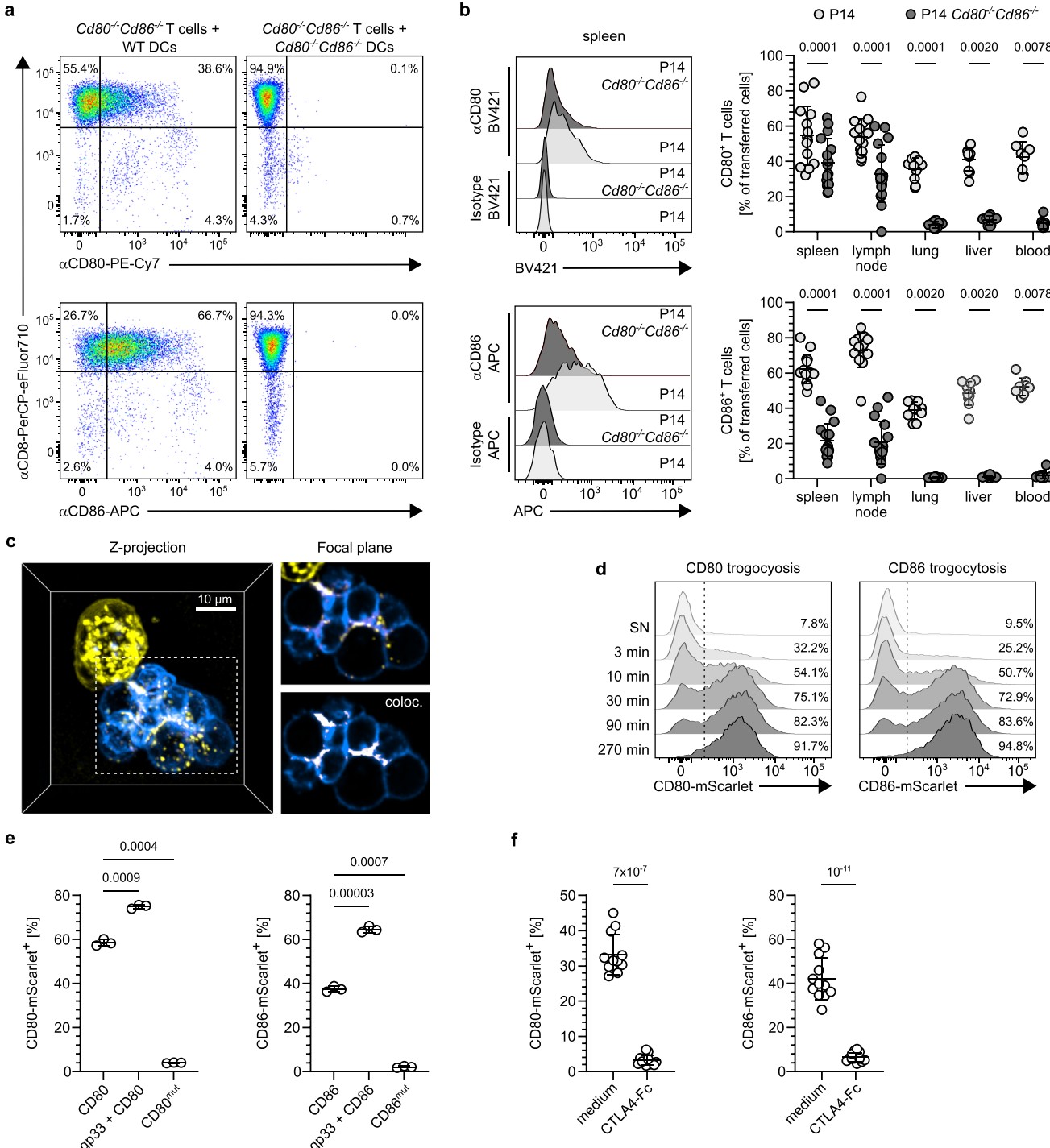

**Fig. 1 | T cells acquire CD80 and CD86 from neighboring cells. a** CD80- and CD86-surface staining of *Cd80⁻/⁻Cd86⁻/⁻* P14 CD8⁺ T cells co-cultured with wild-type (WT) or *Cd80⁻/⁻Cd86⁻/⁻* antigen-loaded DCs. Gating: Thy1.2⁺ CD44⁺ T cells. **b** CD80 and CD86 staining of WT and *Cd80⁻/⁻Cd86⁻/⁻* P14 T cells on day 12 post LCMV-infection. Left: representative histograms of transferred T cells. Right: Proportion of CD80⁺ or CD86⁺ cells. LN denotes lymph nodes. **c** Z-projection (left) and focal plane (right) of membrane-Tomato P14 CD8⁺ T cells (blue) co-cultured for 48 h with CD80-TagGFP transgenic, antigen-loaded MEFs (yellow). Bottom right image solely depicts CD80 co-localizing with T cell membrane. Scale bar 10 μm. **d** Acquisition kinetics of CD80 and CD86 by P14 CD8⁺ T cells upon co-culture with CD80- or CD86-mScarlet-transgenic, antigen-expressing MEFs or their respective supernatants (SN). **e** Proportions of P14 CD8⁺ T cells acquiring CD80 or CD86 co-cultured with MEFs expressing native CD80- or CD86-mScarlet or mutants thereof (CD80 Y201A, CD86 Q35A) with or without antigen-loading. **f** Blockade of intercellular

transfer of CD80 and CD86 upon addition of CTLA4-Fc to CD8⁺ T cells co-cultured with CD80- or CD86-mScarlet expressing MEFs. See also Figure S1 and Movie S1. Statistics: **b**: Pooled data from 4 independent experiments. Two-sided Wilcoxon matched-pairs signed rank test. *p*-values: CD80 spleen 0.0001 (*n* = 14 animals); CD80 iLN 0.0001 (*n* = 14 animals); CD80 lung 0.0020 (*n* = 10 animals); CD80 liver 0.0020 (*n* = 10 animals); CD80 blood 0.0078 (*n* = 8 animals); CD86 spleen 0.0001 (*n* = 14 animals); CD86 iLN 0.0001 (*n* = 14 animals); CD86 lung 0.0020 (*n* = 10 animals); CD86 liver 0.0020 (*n* = 10 animals); CD86 blood 0.0078 (*n* = 8 animals). **e**: Representative data from 1 of 3 independent experiments. Brown-Forsythe/Welch's ANOVA with Dunnett's T3 correction for multiple comparison. *p*-value: CD86 vs. gp33 + CD86: 3 × 10⁻⁵ (*n* = 3 biologically independent samples). **f**: Pooled data from 3 independent experiments. Two-sided Mann–Whitney test. *p*-values: CD80: 7 × 10⁻⁷ CD86: 1 × 10⁻¹¹ (*n* = 12 biologically independent samples). Source data are provided as a Source data file.

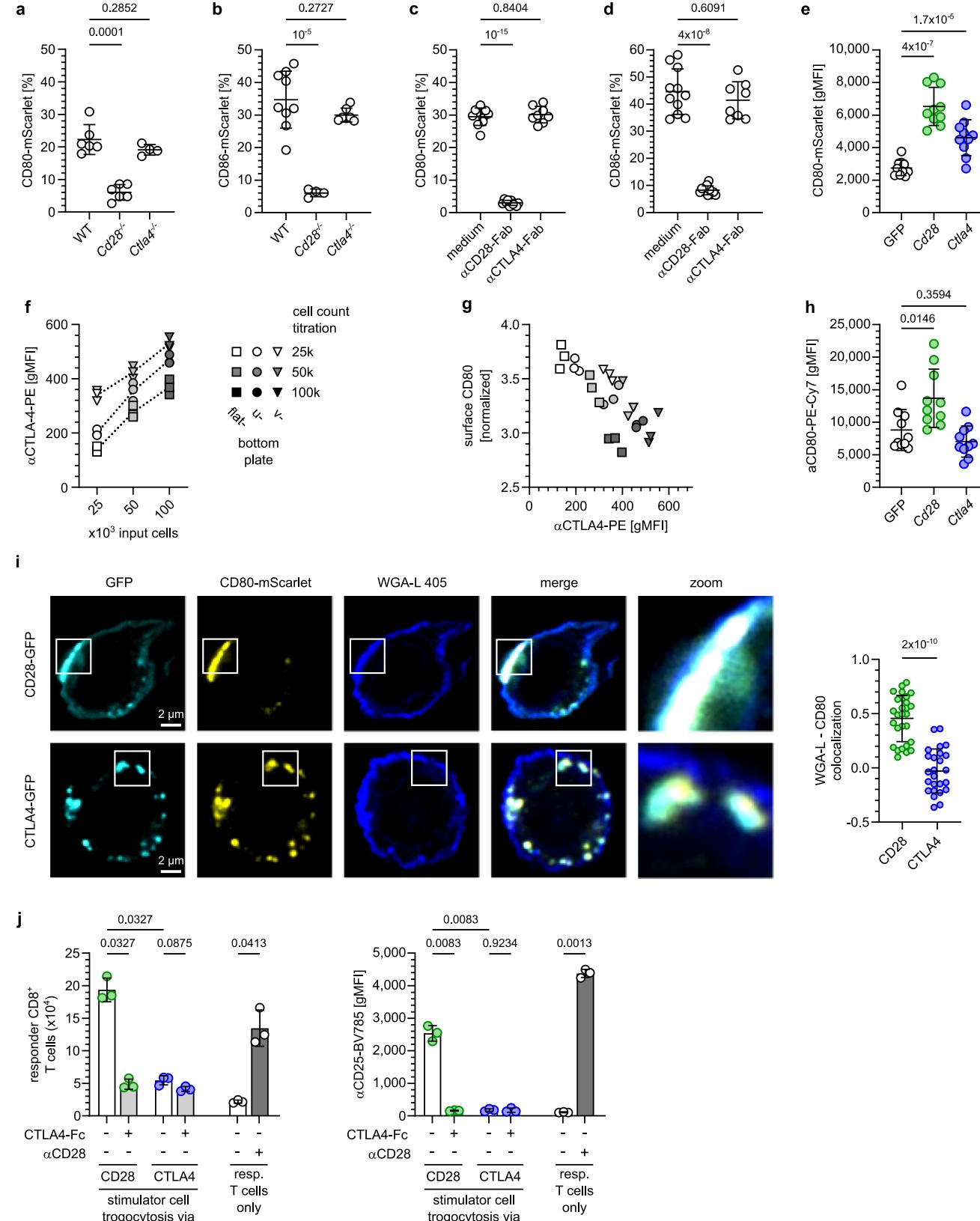

Within the cell clusters forming around activated antigen-presenting cells (APCs) T cells can express both CTLA4 and CD28[5]. When we studied trogocytosis of CD80 by activated $Cd80^{-/-}$ $Cd86^{-/-}$ CD8[+] T cells, we observed that CTLA4 expression scaled positively with density and abundance at which the T cells had been cultured (Fig. 2f), but negatively with the amount of trogocytosed CD80 detectable on

the cell surface (Fig. 2g). These observations may suggest that T cells employ trogocytosis to actively adjust the availability of important co-stimulatory molecules to their population density. Specifically, they lead to the interesting hypothesis that even though both CTLA4 and CD28 mediate trogocytosis, they may confer different fates to trogo-cytosed CD80 molecules with CD28 promoting and CTLA4 inhibiting

**Fig. 2 | CD28 and CTLA4 trogocytose CD80, but direct it towards different fates. a**–**d** Trogocytosis of CD80 (**a**, **c**) or CD86 (**b**, **d**) by naive WT, *Cd28*- or *Ctla4*-/- CD8+ T cells (**a**, **b**) or WT CD8+ T cells upon CD28- or CTLA4-blockade (**c**, **d**). **e** Quantification of CD80 trogocytosed by GFP-, *Cd28*- or *Ctla4*-transduced *Cd80*-/-*Cd86*-/- CD8+ T cells. **f, g** Expression of CTLA4 (**f**) and its correlation with trogocytosed CD80 on T cell surface (**g**) of titrated numbers of *Cd80*-/-*Cd86*-/- P14 CD8+ T cells activated in flat-, u-, or v-bottom plates for 28 h. To correct for different trogocytosis rates, surface CD80 levels were normalized to total acquired CD80-TagRFP (αCD80-PE-Cy7/TagRFP). **h** Quantification of trogocytosed CD80 on surface of *GFP*-, *Cd28*- or *Ctla4*-transduced *Cd80*-/-*Cd86*-/- CD8+ T cells. Note that only CD28-, but not CTLA4-overexpression in T cells increases CD80-surface expression. **i** Images and quantification of co-localization of wheat germ agglutinin lectin (WGA-L) surface staining with CD80 trogocytosed by CD28- or CTLA4-expressing 58αβ T cells. Scale bar 2 μm. **j** Trogocytosis of CD80 via CD28 but not via CTLA4 enables co-stimulation of neighboring cells. *Cd28*- and *Ctla4*-transduced *Cd80*-/-*Cd86*-/- B cells were co-cultured with CD80-mScarlet expressing MEFs, FACS-sorted for mScarlet-expression, gp33-antigen-loaded and co-cultured with *Cd80*-/-*Cd86*-/- P14 CD8+ T cells with or without addition of CTLA4-Fc ("specificity control") or agonistic αCD28 antibodies ("positive control"). Plots depict day 3 T cell counts (left) and CD25 expression (right). See also Figure S2. Statistics: **a**: Pooled data from 3 independent experiments. Brown-Forsythe/Welch's ANOVA with Dunnett's T3 correction. WT and *Cd28*-/- (*n* = 6 biologically independent samples), *Ctla4*-/- (*n* = 4 biologically independent samples). **b**: Pooled data from 3 independent experiments. Brown-Forsythe/Welch's ANOVA with Dunnett's T3

correction. *p*-values: WT vs. *Cd28*-/-: 0.00001; WT vs. *Ctla4*-/-: 0.27 WT (*n* = 9 biologically independent samples), *Cd28*-/- (*n* = 4 biologically independent samples), *Ctla4*-/- (*n* = 7 biologically independent samples). **c**: Pooled data from 3 independent experiments. Brown-Forsythe/Welch's ANOVA with Dunnett's T3 correction. *p*-values: medium vs. α-CD28-Fab: $1 \times 10^{-15}$; medium vs. α-CTLA4-Fab: 0.84 (medium and α-CD28-Fab: *n* = 10 biologically independent samples, α-CTLA4-Fab: *n* = 8 biologically independent samples). **d**: Pooled data from 3 independent experiments. Brown-Forsythe/Welch's ANOVA with Dunnett's T3 correction. *p*-values: medium vs. α-CD28-Fab: $4 \times 10^{-8}$; medium vs. α-CTLA4-Fab: 0.61. (medium and α-CD28-Fab: *n* = 11 biologically independent samples, α-CTLA4-Fab: *n* = 8 biologically independent samples). **e**: Pooled data from 3 independent experiments. Repeated-measures one-way ANOVA with Dunnett's T3 correction. *p*-values: GFP vs. Cd28: $4 \times 10^{-7}$; GFP vs. Ctla4: 0.000017 (*n* = 10 biologically independent samples per group). **f**: Representative data from 1 of 2 independent experiments. (*n* = 3 biologically independent samples per group). **g**: Representative data from 1 of 2 independent experiments (*n* = 3 biologically independent samples per group). **h**: Pooled data from 3 independent experiments. Friedmann test with Dunn's correction. *p*-values: GFP vs. Cd28: 0.015; GFP vs. Ctla4: *p* value: 0.36 (*n* = 10 biologically independent samples per group). **i**: Two-sided Mann–Whitney test; *p*-value: $2 \times 10^{-10}$ (CD28: *n* = 28 cells and CTLA4: *n* = 25 cells, examined over 2 independent experiments). **j**: Representative data from 1 of 3 independent experiments. Repeated-measures one-way ANOVA with Holm-Šídák's correction (*n* = 3 biologically independent samples per group). Source data are provided as a Source data file.

their surface expression. Consistent with this hypothesis, we found that while overexpressing CD28 or CTLA4 in T cells both increased trogocytosis of CD80 (Fig. 2e), only for CD28 this was accompanied by an increase in surface staining by Anti-CD80-antibodies (Fig. 2h). This was also corroborated in T cell lines engineered to express either CD28 or CTLA4 (Supplementary Fig. 2c). These differences could indicate that CD28 and CTLA4 differ in their ability to direct CD80 to the cell surface or that *cis*-interactions of CTLA4 with CD80 on the cell surface impair CD80-staining with antibodies. Attesting to the latter scenario, we found that binding of CTLA4-Fc, but not CD28-Fc, to CD80 impaired subsequent Anti-CD80 antibody staining (Supplementary Fig. 2d). This effect was also observed upon co-expressing CD80 and a cell-surface-targeted CTLA4 variant (Supplementary Fig. 2e). This indicates that *cis*-interactions between CD80 and CTLA4 at the cell surface sterically hinder Anti-CD80 antibody staining, thereby rendering this approach unreliable for comparing cell surface located CD80 upon CD28- versus CTLA4-mediated trogocytosis. To circumvent this pitfall, we determined the location of trogocytosed CD80 by confocal microscopy. We found more fluorescently-tagged CD80 co-localizing with the cell surface upon trogocytosis by CD28 vs. CTLA4 (Fig. 2i), corroborating that CD28 more efficiently directs trogocytosed CD80 to the cell surface.

We then tested the functionality of trogocytosed CD80 located on the recipient cell surface. For this, we flow-cytometrically sorted activated P14 *Cd80*-/-*Cd86*-/- T cells that had trogocytosed CD80-mScarlet, and co-cultured them with naive, CFSE-labeled P14 *Cd80*-/-*Cd86*-/- T cells in presence of gp33-antigen. As both cell populations are on a *Cd80*-/-*Cd86*-/- background any CD80 can solely derive from trogocytosed molecules. Since naive T cells require concomitant TCR- and CD28-ligation, full T cell activation only occurs if trogocytosed CD80 is located at the cell surface and not blocked by cis-interactions. We observed that T cells that had trogocytosed CD80 efficiently activated naive T cells (Supplementary Fig. 2f). Potential confounders of this experimental setup are TCR-mediated bystander trogocytosis, which may affect the trafficking of trogocytosed CD80 and TCR-mediated production of IL-2 and other cytokines. Furthermore, the non-exclusive expression of CD28 and CTLA4 by *Cd80*-/-*Cd86*-/- T cells does not allow to distinguish CD28- and CTLA4-mediated effects. To overcome these limitations, we modified the above experimental setup. For this we took into account that engineered expression of CD28 or CTLA4 was both sufficient and necessary to enable different

cell types to trogocytose CD80 (Fig. 2e, Supplementary Fig. 2a–c)[19,20]. We transduced primary *Cd80*-/-*Cd86*-/- B cells, which lack endogenous expression of CD28 and CTLA4, with CD28 or CTLA4, let them trogocytose CD80-mScarlet, loaded them with antigen and co-cultured them with naive, CFSE-labeled P14 *Cd80*-/-*Cd86*-/- T cells. We observed that only cells trogocytosing CD80 via CD28, but not via CTLA4, efficiently activated naive T cells (Fig. 2j). Together these experiments confirm that CD28 and CTLA4 direct trogocytosed CD80 to different fates, that CD80 trogocytosed by CD28 is presented in a functional manner at the cell surface and that T cells can utilize trogocytosed CD80 to activate neighboring naive T cells.

The different functional outcomes of CD28- versus CTLA4-mediated trogocytosis could be explained by at least two different mechanisms possibly acting in concert: First, CD28 more efficiently directs acquired ligands to the cell surface than CTLA4. Second, those acquired ligands that locate to the cell surface upon CTLA4-mediated trogocytosis are functionally silenced by cis-interactions and, hence, unavailable for stimulating CD28 on neighboring cells. These results have important implications for understanding collective regulation of T cells clustering around APCs, because CD8+ T cells express CD28 constitutively, but CTLA4 only more than 24 h after activation and in a cell-density dependent manner (Fig. 2g)[5]. Hence, early after activation when T cell density is low and few of them express CTLA4, trogocytosis promotes mutual stimulation of clustering T cells, whereas it becomes more and more inhibitory as T cells proliferate, which increases their density and drives CTLA4 expression.

## CD28 and CTLA4 direct acquired CD80 to different compartments

Next we set out to dissect the different fates of ligands trogocytosed via CD28 versus CTLA4. Studies using antibodies against CD28 and CTLA4 have shown that both receptors can undergo endocytosis[21–24]. Accordingly, we observed ligands acquired via CTLA4 and CD28 in delimited intracellular spots, consistent with an endosomal location (Fig. 3a). To not solely rely on imaging studies, we developed a functional assay reporting on the fate of trogocytosed ligands. For this, we exploited that distinct endosomal compartments differ in their luminal acidity and determined the pH-levels experienced by ligands trogocytosed via CD28 versus CTLA4. Specifically, we created a ratiometric pH-reporter by attaching pH-sensitive pHluorin to the extracellular and pH-stable TagRFP to the intracellular domain of CD80 (Fig. 3b). Control

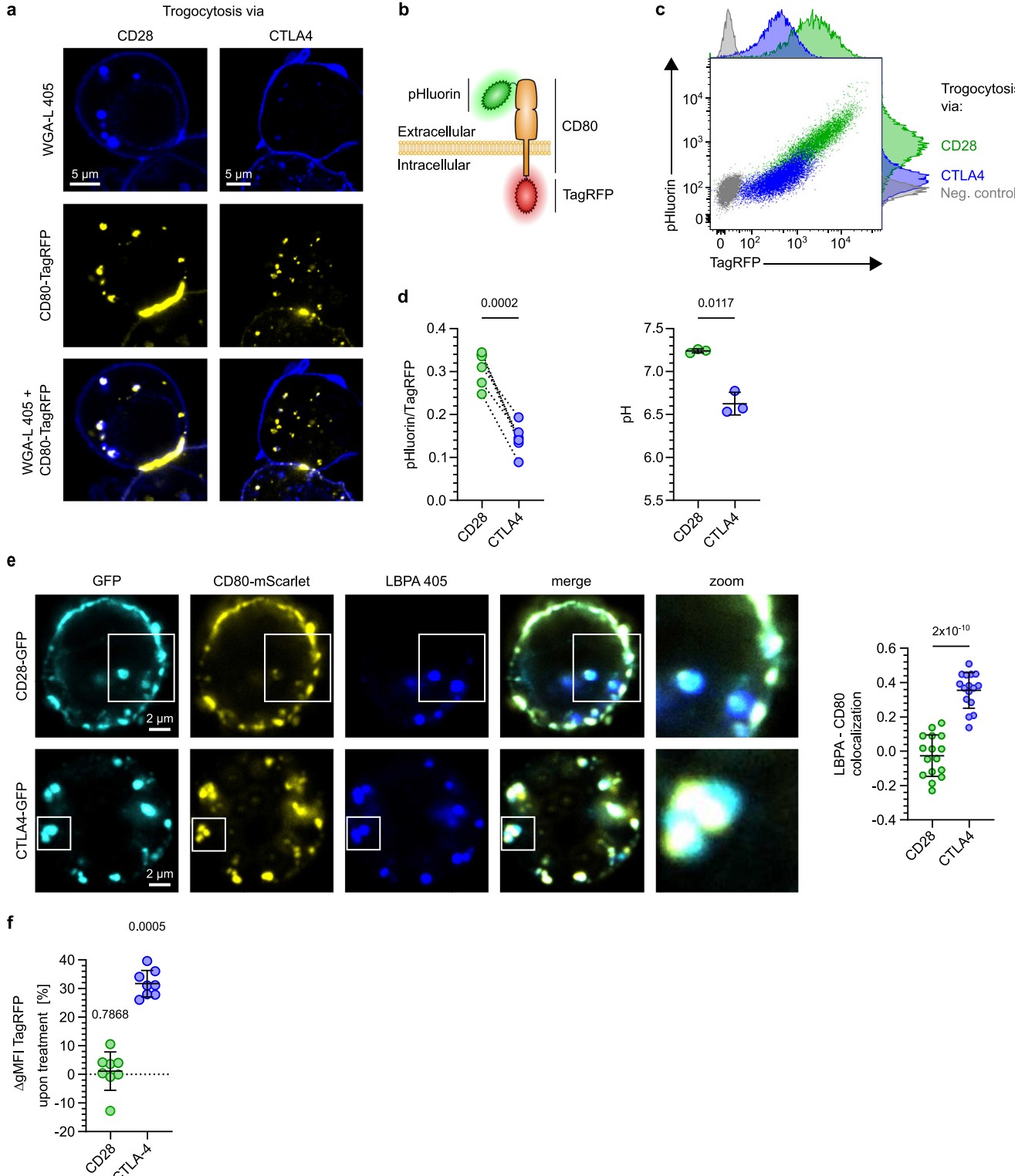

experiments revealed that the ratio of both fluorescent signals mirrored ambient pH-levels (Supplementary Fig. 3a–c). Despite binding CD28 and CTLA4 less well than native CD80 (Supplementary Fig. 3d), this construct was still efficiently trogocytosed (Fig. 3c). We observed a significantly lower pHluorin/TagRFP-ratio and, accordingly, also pH-level for CD80 acquired via CTLA4 vs. CD28 (Fig. 3c, d). This indicates that CTLA4 targets acquired ligands to more acidic compartments than CD28. Consistently, upon CTLA4-mediated trogocytosis we observed a stronger co-localization between a marker for acidic late endosomes (lysobisphosphatidic acid (LBPA)) and CD80 (Fig. 3e).

Protein trafficking to late-endosomes is not tantamount to their subsequent degradation in lysosomes[25]. Alternatively, proteins may still recycle to the cell surface. Thus, we set out to quantify lysosomal degradation of CD80 trogocytosed via CTLA4 and CD28. As the activity of lysosomal proteases is optimal at acidic and impaired at neutral pH[26], an increase in CD80 abundance upon inhibition of lysosomal acidification by Bafilomyin mirrors the decrease in its degradation. We quantified the amount of trogocytosed CD80 molecules by an N-terminal TagRFP-tag, which—due to its low pKa—tolerates endo-/lysosomal pH with little quenching[27]. For CTLA4- and CD28-mediated

**Fig. 3 | CD28 and CTLA4 target trogocytosed CD80 to different cellular compartments. a** Images of trogocytosed CD80-TagRFP acquired by CD28- or CTLA4-TagGFP-expressing 58αβ A2 T cells, whose cell-surface and endosomal membranes were stained with wheat germ agglutinin lectin (WGA-L). Scale bar 5 μm. **b–d** Trogocytosed CD80 experiences different pH-levels upon acquisition via CD28 versus CTLA4. **b** Cartoon depicting design of pHluorin-CD80-mScarlet reporter. **c, d** CD28- or CTLA4-expressing 58αβ A2 T cells were co-cultured with CHO cells expressing pHluorin-CD80-TagRFP. Representative dot plots depicting pHluorin and TagRFP signals of T cells (**c**), quantification of pHluorin/TagRFP ratio (**d**, left) and extrapolation of pH-levels experienced by trogocytosed CD80 (**d**, right) based on standard curve shown in Figure S3. **e** Images and quantification of co-localization of late lysobisphosphatidic acid (LBPA) (late endosome) staining with CD80 trogocytosed by CD28- or CTLA4-TagGFP expressing 58αβ T cells. Scale bar

2 μm. **f** CTLA4 is more efficient in targeting trogocytosed CD80 for degradation than CD28. CD28 or CTLA4 expressing 58αβ A2 T cells were co-cultured with CHO cells expressing CD80-TagRFP-pHluorin with or without Bafilomycin. Plot depicts increase of TagRFP gMFI upon Bafilomycin treatment. See also Figure S3. Statistics: **d**: pHluorin/TagRFP: pooled data from 6 independent experiments. Two-sided paired $t$ test ($n = 6$ biologically independent samples). pH: pooled data from 3 independent experiments. Two-sided paired $t$ test ($n = 3$ biologically independent samples). **e**: Two-sided unpaired $t$ test with Welch's correction; $p$-value: $2 \times 10^{-10}$ ($n = 16$ cells per group examined over 2 independent experiments). **f**: Pooled data from 8 independent experiments. Two-way repeated-measure ANOVA with Šídák's correction, treated vs untreated ($n = 8$ biologically independent samples per group). Source data are provided as a Source data file.

trogocytosis, inhibition of lysosomal function increased TagRFP signals by approximately 30% and 1%, respectively (Fig. 3f). Taken together these experiments demonstrate that CTLA4 targets trogocytosed CD80 to more acidic, late endosomes and more efficiently directs them towards lysosomal degradation than CD28.

## Dissecting the mechanisms of trogocytosis via CD28 and CTLA4

Conceptually, trogocytosis may involve endocytosis of donor cell membrane-derived vesicles (ectosomes), thereby forming intraluminal vesicles in recipient cell endosomes (Fig. 4a, top). Alternatively, trogocytosis may involve transient fusions of donor and recipient cell membranes (Fig. 4a, bottom). In both scenarios the location of the intracellular (IC) domains of trogocytosed ligands differs. Specifically, in the ILV scenario their IC domains locate to the endosomal lumen, whereas in the membrane fusion scenario they reside in the recipient cell cytosol. As endosomes are more acidic than the cytosol, measuring the ambient pH-levels experienced by the intracellular domain of trogocytosed ligands might allow to distinguish between both scenarios. For this, we modified the ratiometric pH-reporter by attaching both fluorochromes to the CD80 IC domain (Supplementary Fig. 3e). We observed a significantly lower ambient pH of the CD80 IC domain upon trogocytosis via CTLA4 vs. CD28, which was also below the cytosolic pH-range of 7–7.4 reported for lymphocytes (Fig. 4b)[28–31]. Hence, at least for CTLA4-mediated trogocytosis the data support the scenario of acquired ligands locating to ILVs within acidic endosomes.

For CD28-mediated trogocytosis, the relatively high pH-level experienced by the intracellular domain does not permit such a conclusion: it could either be explained by a membrane fusion scenario or by targeting to vesicles located in a non-acidic compartment, e.g., at the cell surface. To distinguish between these scenarios, we developed a bi-molecular fluorescence complementation assay reporting on membrane fusion events, irrespective of whether they occur at the cell surface or within endosomes (Fig. 4c). For the assay, we linked a non-fluorescent GFP11-subunit to the intracellular domain of CD80-mScarlet and expressed it in donor cells. Furthermore, we expressed a non-fluorescent GFP1–10 subunit in the cytosol of T lymphocytes. Upon fusion of donor and recipient cell membranes, GFP11-subunits would translocate to the cytosol of the T cell, where they encounter GFP1–10 subunits. Since GFP1–10 and GFP11 subunits spontaneously self-assemble into fluorescent GFP chromophores[32], fusion of donor and recipient membranes upon trogocytosis should thus yield a green fluorescent signal. Indeed, positive control samples became bright green (Fig. 4d). In contrast, there was only a weak green signal detectable in GFP1–10 expressing recipient T cells that had trogocytosed CD80-mScarlet-GFP11 (detectable by the mScarlet-derived red fluorescence) (Fig. 4d). Likewise, when we performed the assay using GFP1–10-expressing T cell lines expressing CD28 or CTLA4, there was only a weak, albeit reproducible and statistically significant green signal (Fig. 4e). These results argue that trogocytosis via both CD28 and CTLA4 involves only very little membrane fusion, supporting the notion that acquired ligands are mainly located on membrane vesicles.

An open question is how we can reconcile the substantial amounts of trogocytosed CD80 on the cell surface with only very little membrane fusion. A nearby explanation would be that intraluminal vesicles arising upon trogocytosis became extracellular vesicles upon recycling to the cell surface. If such CD80-expressing vesicles remained at least in part attached to the cell surface (e.g., by interaction with CD28 on the cell surface), they would be detectable by Anti-CD80 antibodies and could provide co-stimulation to neighboring T cells. If this was indeed the case, proteolytic cleavage of CD80 or CD28 should release the extracellular vesicles, resulting in cells losing the fluorescent signal attached to the CD80 IC domain (Supplementary Fig. 4a, left). In contrast, if trogocytosed CD80 molecules were integrated into the recipient cell membrane, then the fluorescent signal attached to their intracellular domain should be unaffected by proteolytic cleavage of CD80 (Supplementary Fig. 4a, right). As both native CD28 and CD80 were highly resistant to proteases (Supplementary Fig. 4b), we rendered CD80 protease-cleavable by engineering an IgG-domain into its stalk region (Supplementary Fig. 4b). Exposing cells that had trogocytosed CD80-IgG via CD28 to trypsin markedly reduced the mScarlet-derived signal attached to the CD80 IC domain (Fig. 4f). This suggests that a substantial fraction of trogocytosed CD80 was not integrated into the recipient cell membrane, but rather located to extracellular vesicles at the cell surface.

## Receptor EC- and IC-domains determine ligand fate

Fates of endocytosed ligands can be determined by the receptor intra-(IC) and extracellular (EC) ligand-binding domain[33,34]. As CD28 and CTLA4 confer divergent fates to trogocytosed ligands, we set out to investigate the contributions of their IC and EC domains. For this, we generated hybrid variants of CD28 and CTLA4 with swapped IC domains (Fig. 5a). When we compared the efficacy of trogocytosis across native and hybrid receptors, we noted that only 10% of FACS-sorted cells expressing the CD28/CTLA4(IC) hybrid receptor had trogocytosed CD80-TagRFP compared to more than 80% of cells expressing CD28, CTLA4 or the CTLA4/CD28(IC) hybrid receptor (Fig. 5b). Among cells that trogocytosed CD80, those expressing CD28 or CTLA4/CD28(IC) harbored substantially more CD80 molecules than those expressing CTLA4 or the CD28/CTLA4(IC) hybrid, as indicated by the brighter TagRFP-fluorescence intensity (Fig. 5c). The lower levels of trogocytosed CD80 in cells expressing receptors containing the CTLA4 IC domain may either be due to less efficient acquisition or to more efficient degradation. To quantify ligand degradation we determined the increase of CD80-TagRFP signal intensity upon inhibiting lysosomal function. This more profoundly increased the abundance of CD80 trogocytosed by CTLA4 or CTLA4/CD28(IC) compared to CD28 or CD28/CTLA4(IC) (Fig. 5d). Hence, the effect of Bafilomycin on CD80 abundance did not segregate with the intracellular domains of CD28 and CTLA4, but rather with their extracellular domains. In contrast, the pH-level experienced by trogocytosed CD80 segregated with the CD28 and CTLA4 intracellular domains, with CD80 locating to more acidic compartments upon trogocytosis via CTLA4 and CD28/CTLA4(IC) than

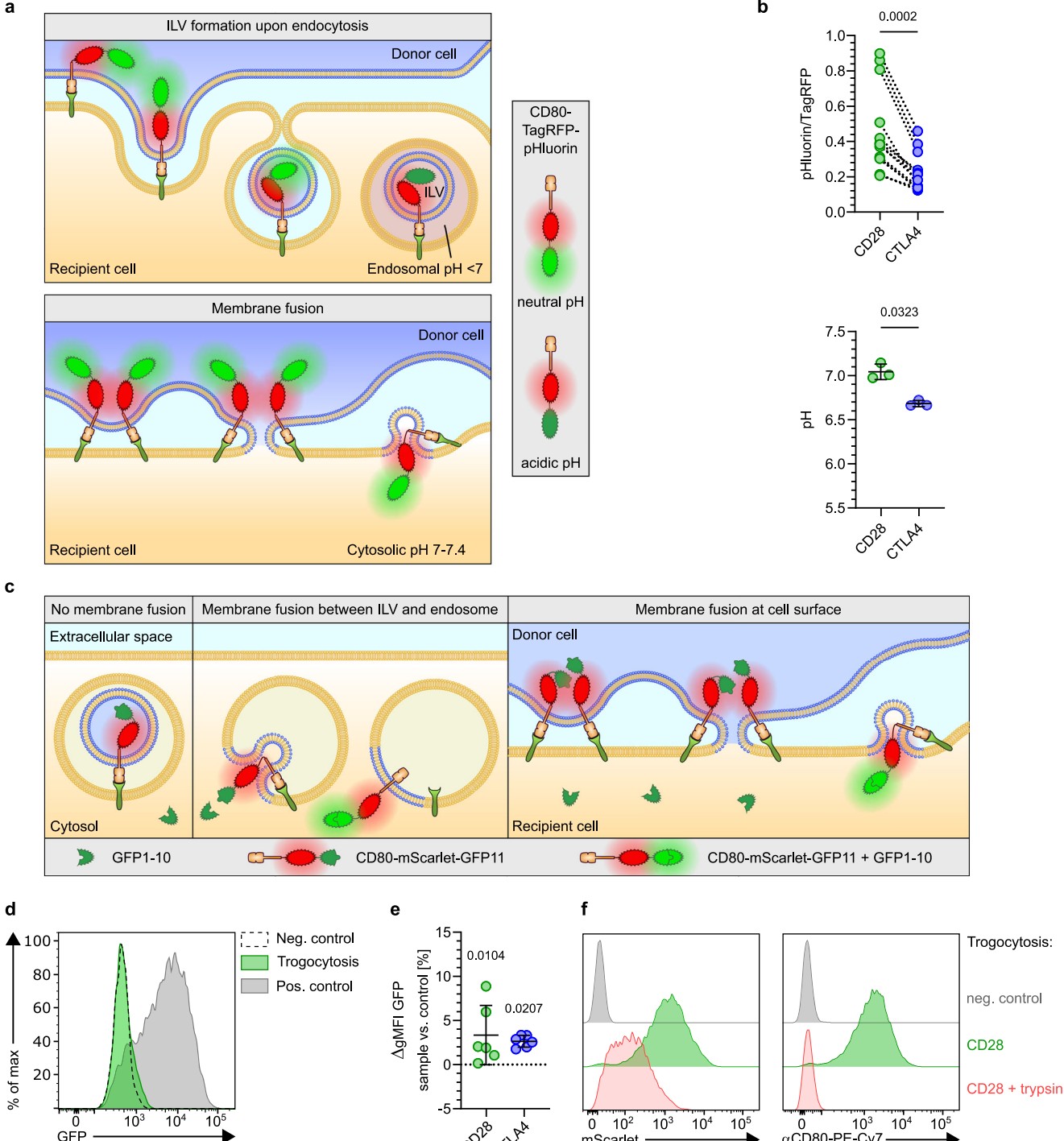

**Fig. 4 | Dissecting mechanisms underlying trogocytosis. a** Cartoons depicting putative mechanisms of trogocytosis via intraluminal vesicle (ILV) generation or membrane fusion between donor and recipient cells. **b** CD80 experiences different pH-levels upon trogocytosis via CD28 versus CTLA4. CD28- or CTLA4-expressing 58αβ A2 T cells were co-cultured with CD80-TagRFP-pHluorin expressing CHO cells. Quantification of pHluorin/TagRFP ratio and extrapolation of pH-levels experienced by trogocytosed CD80 based on standard curve shown in Figure S3. **c**, **d** Analysis of membrane fusion by bi-molecular fluorescence complementation (BIFC) assay. **c** Cartoon depicting putative outcomes of BIFC trogocytosis assay. CD8+ T cells (**d**) or CD28- or CTLA4-expressing 58αβ A2 T cells (**e**) expressing cytosolic GFP1–10 were co-cultured with CD80-mScarlet-GFP11-expressing CHO cells. Positive control: GFP1–10 transgenic T cells transduced with CD80-mScarlet-GFP11 (**d**). Negative control: GFP1–10 transgenic T cells cultured with CD80-mScarlet-transgenic CHO cells. Plot shows increase of GFP gMFI over negative control after 24 h (**e**). **f** Fluorescent tag attached to CD80 cytoplasmic domain is cleavable by extracellular protease treatment. CD28-expressing 58αβ A2 T cells having trogocytosed CD80-IgG-mScarlet were treated with or without trypsin. Negative control: T cells cultured without CHO cells. Representative flow cytometry plots depicting loss of trogocytosed mScarlet and cell surface CD80 signal upon trypsinization-treatment. Note that loss of both signals marks the extracellular location of the cytoplasmic domain of trogocytosed CD80. See also Figures S3 and S4. Statistics: **b**: pHluorin/TagRFP: pooled data from 13 independent experiments. Two-sided Wilcoxon matched-pairs signed rank test ($n = 13$ biologically independent samples per group) pH: two-sided paired $t$ test, pooled data from 3 independent experiments ($n = 3$ biologically independent samples per group). 4e: Pooled data from 3 independent experiments. Two-way repeated-measure ANOVA with Šídák's correction, sample vs control ($n = 6$ biologically independent samples per group). Source data are provided as a Source data file.

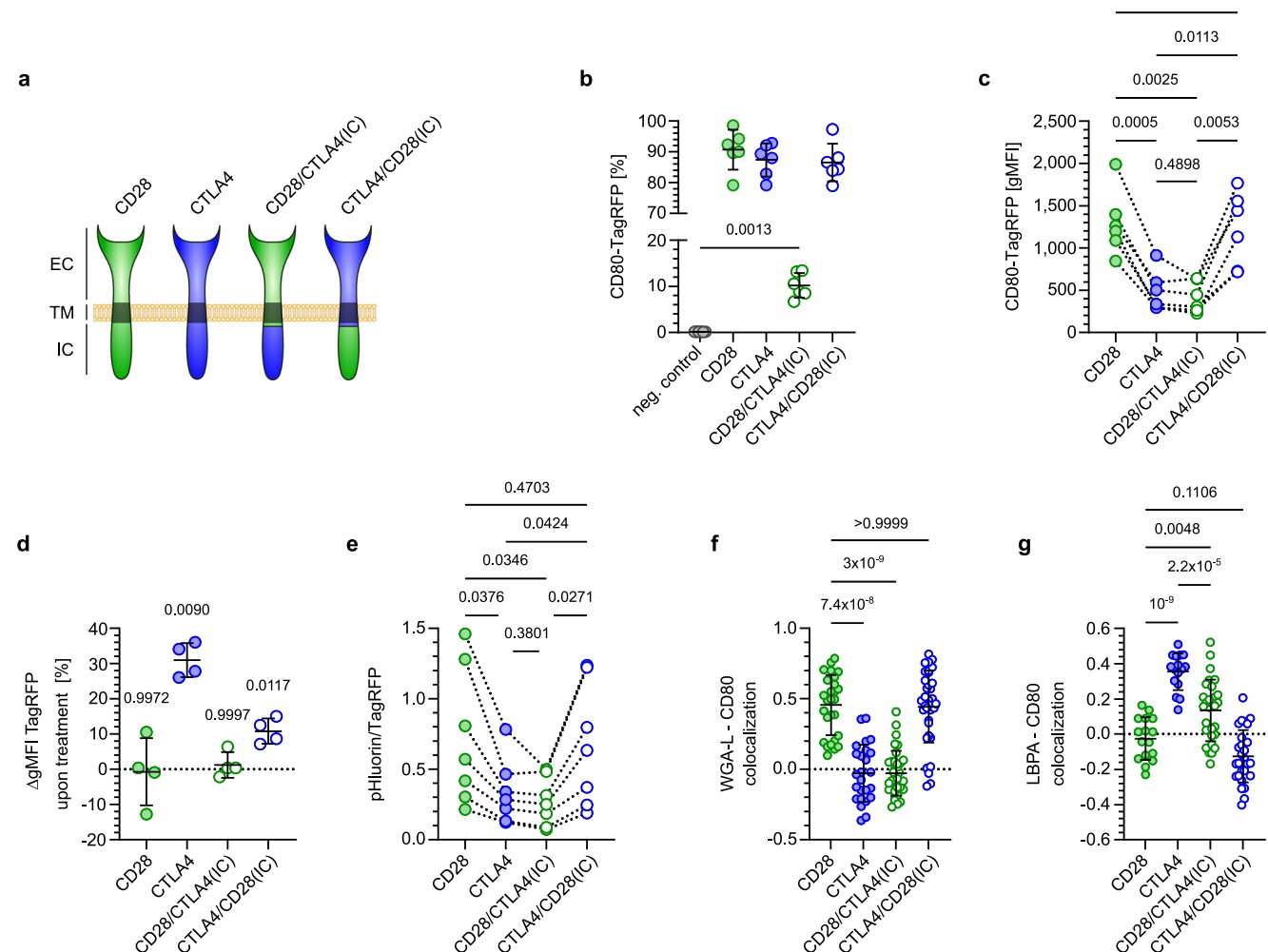

**Fig. 5 | Extra- and intracellular domains of CD28 and CTLA4 modulate fate of trogocytosed CD80. a** Composition of CD28/CTLA4(IC) or CTLA4/CD28(IC) hybrid receptors compared to native CD28 and CTLA4. **b**–**g** 58αβ A2 T cells expressing native or hybrid receptors co-cultured with CHO cells expressing CD80-TagRFP-pHluorin were analyzed by flow-cytometry (**b**–**e**) or confocal microscopy (**f**, **g**). Proportion of T cells acquiring CD80 (**b**) and amount of trogocytosed CD80 detectable per acquiring cell (**c**). Note that while only few T cells expressing the CD28/CTLA4(IC) hybrid receptor trogocytose CD80 (shown in **b**), the amount per acquiring cell is comparable to T cells expressing native CTLA4 (shown in **c**). **d** Increase in TagRFP signal intensity (gMFI) upon addition of Bafilomycin during trogocytosis of CD80-TagRFP. **e** pHluorin/TagRFP ratio of CD80-TagRFP-pHluorin trogocytosed via different receptors. Note that while receptors containing the CTLA4 <u>intracellular</u> domain target trogocytosed CD80 to more acidic compartments (shown in **e**), blockade of lysosomal function only affects CD80 trogocytosed by receptors containing the CTLA4 <u>extracellular</u> domain (shown in **d**). Quantification of co-localization of trogocytosed CD80 with wheat germ agglutinin lectin (WGA-L) (**f**) or lysobisphosphatidic acid (LBPA) (**g**). Statistics: **b**, **c**: Pooled data from 6 independent experiments. Repeated measures one-way ANOVA with Tukey's correction ($n = 6$ biologically independent samples per group). **d**: Pooled data from 4 independent experiments. Two-way repeated-measure ANOVA with Šídák's, treated vs untreated ($n = 4$ biologically independent samples per group). **e**: Pooled data from 7 independent experiments. Repeated measures one-way ANOVA with Tukey's correction ($n = 7$ biologically independent samples per group). **f**: Kruskal-Wallis test with Dunn's correction: $p$-values: CD28 vs. CTLA4: $7.4 \times 10^{-8}$; CD28 vs. CD28/CTLA4(IC): $3 \times 10^{-9}$; CD28 vs. CTLA4/CD28(IC): 0.99. (CD28: $n = 28$ cells, CTLA4: $n = 25$ cells, CD28/CTLA4(IC) and CTLA4/CD28(IC): $n = 35$ cells each, all examined over 2 independent experiments). **g**: Pooled data from two independent experiments. Brown-Forsythe/Welch's ANOVA with Dunnett's T3 correction. $p$-values: CD28 vs. CTLA4: $1 \times 10^{-9}$; CD28 vs. CD28/CTLA4(IC): 0.0048; CD28 vs. CTLA4/CD28(IC): 0.11; CTLA4 vs. CD28/CTLA4(IC): $2.22 \times 10^{-5}$ (CD28 and CTLA4: $n = 16$ cells each, CD28/CTLA4(IC) and CTLA4/CD28(IC): $n = 29$ cells each, all examined over 2 independent experiments). Source data are provided as a Source data file.

via CD28 and CTLA4/CD28(IC) (Fig. 5e). Accordingly, CD80 trogocytosed via CD28 and CTLA4/CD28(IC) mainly co-localized with the cell membrane, whereas those acquired via CTLA4 and CD28/CTLA4(IC) co-localized with LBPA-expressing late endosomes (Fig. 5f, g). Together these results reveal that while the receptor IC domain directs trogocytosed ligands to different compartments, the receptor EC domain plays an important role in determining whether they become degraded.

## Acidification destabilizes CD80 binding to CD28, but not to CTLA4

What mechanism could underlie the role of the receptor EC domains in ligand-degradation? Upon endocytosis the ligand-binding receptor EC

domains move from the neutral extracellular space to the acidic endosomal lumen. Such pH-changes can alter electrostatic interactions and protein conformation, both of which are critical for receptor-ligand interactions. The isoelectric points of the CD28 and CTLA4 extracellular domains (8.6 and 4.5, respectively) indicate that in this pH-range both receptors have opposite net charges. Furthermore, endosomal acidification in- and decreases the net charges of CD28 and CTLA4, respectively. Lower charges enable receptors to better accommodate charge repulsions exerted by ligands, suggesting that endosomal acidification stabilizes the binding of CD80 to CTLA4, but destabilizes its binding to CD28. In other systems, it has been found that dissociation of receptor and ligand upon endosomal acidification

promotes their recycling to the cell surface, whereas acid-stable interaction drives their lysosomal degradation[33,35,36]. This fits well to our observation that interaction of CD80 with the EC domain of CTLA4, but not CD28, promotes its degradation (Fig. 5d).

We mapped acid-sensitive regions of the CD28 and CTLA4 extracellular domains by predicting pH-induced charge alterations of amino acid sets (Supplementary Figs. 5, 6). This revealed two acid-sensitive regions for CD28 of all species analyzed, whereas CTLA4 had either one or none of such regions. One of the two acid-sensitive regions of CD28 was evolutionary conserved (Supplementary Fig. 5), whereas the corresponding region of CTLA4 was acid-resistant across species (Supplementary Fig. 6). Amino acid charges can be modulated by pH-dependent reversible (de-) protonation. This is best exemplified by histidine, whose state switches from non-protonated, neutral at pH 7 to protonated, charged at endosomal pH. In CD28 of different species histidines are dispersed across the EC domain, two of which are evolutionarily conserved (Supplementary Fig. 7). In contrast, the CTLA4 EC domain is largely devoid of histidines, with those present being non-conserved (Supplementary Fig. 7).

To visualize acid-sensitive regions of CD28 and CTLA4, we overlaid their published structures with the differences in electrostatic potential (ΔE) revealed by solving the Poisson-Boltzmann equations at pH 7 and 5 (Fig. 6a, b). As expected, ΔE centered around histidines, but did not include effects of CD28 H139 as this region was not solved in the crystal structure and, hence, could not be included into this analysis. Nevertheless, embedding ΔE visualization into the complex formed by modeled homodimers of each receptor with CD80 suggested that it affects the receptor-ligand interaction and the homodimer region of human CD28, but not of CTLA4 (Fig. 6a, b, Supplementary Movie 2).

First we investigated effects of acidification on the receptor-ligand interface using molecular dynamic (MD) simulations of the interaction of monomeric CD28 and CTLA4 with the receptor-binding domain of CD80. The simulations predicted stable interactions of CD28 with CD80 at pH 7, but a reorientation of the ligand-interaction site at pH 5, suggesting a loss in stability (Fig. 6c, Supplementary Fig. 8a). In contrast, for CTLA4 we found stable ligand-interactions at both pH-levels (Fig. 6d, Supplementary Fig. 8a). To better visualize these effects we depicted the tips of normal vectors of CD28 and CTLA4 and found very similar patterns for CTLA4 at neutral and acidic pH, but a clear shift of CD28 upon acidification (Fig. 6e, Supplementary Fig. 8f). We then analyzed the strength of monomeric receptor-ligand interactions at neutral and acidic pH by umbrella sampling simulations. This showed that acidification decreased the energy required to dissociate CD80 and CD28, whereas it had an opposite effect on CTLA4 (Fig. 6f). Together our results indicate that acidification stabilizes the monomeric interaction of CD80 with CTLA4, but destabilizes its binding to CD28.

Since we also identified an acid-sensitive site at the CD28 homodimer region centering around two evolutionarily conserved histidines (Supplementary Fig. 7), we next analyzed the impact of endosomal pH on this region. MD simulations of CD28 homodimers indicated that acidification increased the flexibility of both monomers without promoting a specific conformation, thereby destabilizing the homodimer configuration (Fig. 6g, Supplementary Fig. 8b). We set out to test these simulations in wet-lab experiments by modifying the CD28 homodimer conformation in a way that mirrors the effects of acidification. For this, we reasoned that the conserved H135 H140 residues located in the CD28 homodimerization region are the major drivers of acidification-induced effects on this region. Hence, we replaced them by arginine, which mimics protonated histidine in terms of charge and structure. To verify this approach, we first ran further MD simulations, which showed that H135R H140R mutations increased the flexibility of the CD28 homodimer in a way analogous to acidification (Supplementary Fig. 8b). Upon transducing this double arginine mutant in

cells, we detected reduced expression levels at the cell surface compared to native CD28 (Supplementary Fig. 8c, d). This was not due to less efficient transduction or lower mRNA expression as expression levels of the IRES-controlled Thy1.1 reporter were comparable between native and mutant CD28. Furthermore, when we expressed both variants with a GFP-variant fused to their intracellular domains, we found that expression of CD28 H135R H140R led to a more patchy, intracellular expression pattern compared to native CD28 (Supplementary Fig. 8d). As newly synthesized proteins need to pass certain endoplasmic reticulum (ER) checkpoints, before they are released[37], this increased proportion of intracellular location can be taken as an indicator of altered protein conformation. Importantly, the CD28 H135R H140R protein was not misfolded per se, in which case there would have been no CD28 expression at the cell surface. Rather the reduced cell surface expression level fits well to the notion that mutating H135 and H140 to arginine residues increases the flexibility of CD28, thereby decreasing the probability of achieving a conformation which enables efficient ER export. As the mutations are designed to mimic the effect of histidine protonation, this supports the view that acidification promotes conformational flexibility in the CD28 homodimerization region. To further test the effects of acidification on CD28 conformation, we studied CD28 within endosomes. For this we exploited that in the CD28/CTLA4(IC) hybrid receptor the CTLA4-derived IC domain targets the CD28 extracellular and transmembrane domains to acidic endosomes. We expressed CD28/CTLA4(IC)-GFP or CTLA4-GFP fusion proteins and compared the staining intensity with CD80-Fc with or without blockade of endosomal acidification. For each receptor we normalized the CD80-Fc-derived signal to GFP expression to account for differences in protein expression levels (Fig. 6h, Supplementary Fig. 8e). For CTLA4, addition of Bafilomycin did not change the intensity of CD80-Fc-staining (reflected by a ratio of 1), indicating that its conformation is not altered by endosomal acidification. In contrast, Bafilomycin markedly increased CD80-Fc-staining of the CD28/CTLA4(IC) hybrid receptor, supporting the notion that inhibition of acidification reverses conformational changes CD28 undergoes in endosomes. This indicates that endosomal acidification destabilizes the conformation of CD28, but not of CTLA4, and that this impairs ligand-binding. Note that in the wet-lab experiments we purposely used murine CD28, which shares the conserved histidines at the homodimer region with human CD28, but lacks H56 whose protonation directly affects the ligand-binding interface of human CD28. This allows to attribute the observed effects of acidification on CD28 conformation and ligand-binding to the homodimer region. Finally, we directly measured how acidification affects the stability of the receptor-ligand interaction. When we stained cells expressing CD28 or CTLA4 with CD80-Fc and then lowered the pH, we observed a dissociation of CD80 from CD28, but not from CTLA4 (Fig. 6i, j). Taken together our structural modeling and wet lab experiments concordantly indicate that the interaction of CD80 with CD28 is substantially more acid-sensitive than with CTLA4. Within acidified endosomes this makes CD80 a dissociative ligand for CD28, but not for CTLA4, and enables its return to the cell surface. Conversely, our results highlight that the acid-stability of CTLA4 is crucial for its ability to direct trogocytosed ligands toward lysosomal degradation.

## Discussion

Due to their central role in protection against pathogens and malignancies, mechanisms shaping T cell responses are studied extensively. Most studies focus on how cellular production of different regulatory molecules promotes or curtails T cell responses. A frequently neglected aspect of such regulatory proteins is that their availability may not solely depend on the producing cells. For example, cytokine effects can be modulated by soluble receptors acting as decoys or even as agonists, activating cells that would naturally not respond[38]. Focusing on CD28 and CTLA4, which stimulate and inhibit T cells, respectively,

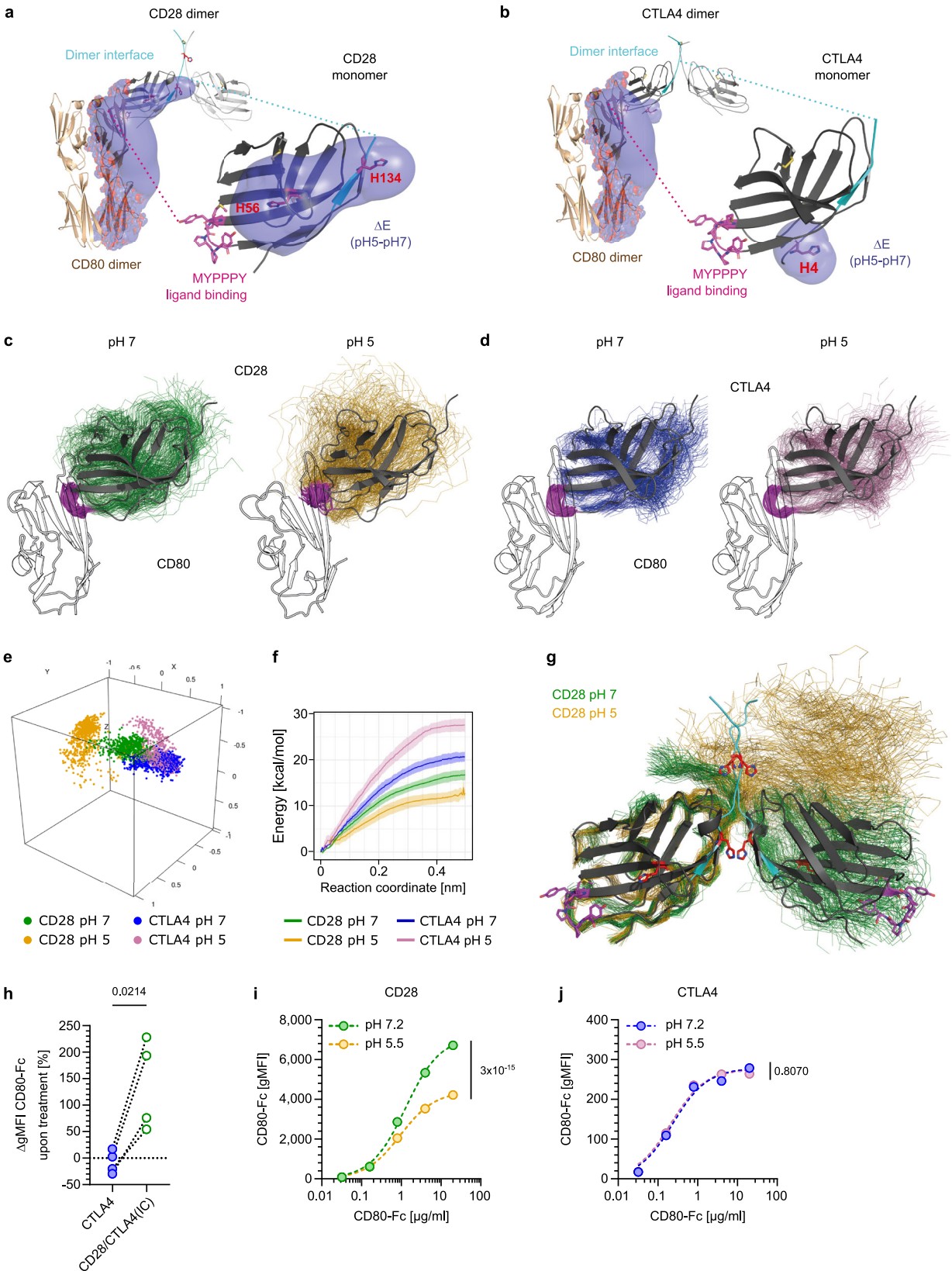

we demonstrated that such non-cell-autonomous regulation of ligand-availability extends to membrane-bound molecules and delineate the underlying principles. Mechanistically, this is achieved by cells extracting ligands from neighboring cells and either re-using or degrading them.

When we compared CD28 and CTLA4, we found that both receptors trogocytose CD80 and CD86 expressed on neighboring cells–consistent with previous publications[15–18]. Conceptually, trogocytosis may involve transient fusions of donor and recipient cell membranes, endocytosis of donor cell membrane-derived ectosomes into recipient

**Fig. 6 | Acidification destabilizes binding of CD80 to CD28, but not to CTLA4.** **a, b** Superimposition of acidification-induced changes in electrostatic potential (ΔE) on monomeric human CD28 or CTLA4 or respective dimers interacting with CD80. Ligand-interaction and homodimer regions of CD28 and CTLA4 are marked in pink and cyan, respectively. Histidines are marked in red. **c, d** Molecular dynamic simulations of monomeric CD28 (C) or CTLA4 (D) interacting with CD80 receptor-binding domain. Ligand interaction sites of CD28 and CTLA4 are highlighted in pink. Note that acidification induces a reorientation of the ligand-interaction site of CD28, but not of CTLA4. **e** Cartesian coordinates of normal vector tips of CD28 and CTLA4 interacting with CD80. **f** Umbrella sampling simulations of monomeric receptor-ligand interactions. Shaded areas depict error estimation. **g** Molecular dynamic simulations of CD28 homodimers at pH 7 vs. 5. Ligand-interaction and homodimer regions are marked in pink and cyan, respectively. Histidines are marked in red. **h** Inhibition of acidification stabilizes endosomal conformation of

CD28. CTLA4-TagGFP or CD28/CTLA4(IC)-TagGFP expressing 58αβ A2 T cells were cultured with or without Bafilomycin. CD80-Fc staining was performed on fixed, permeabilized cells. Plot depicts increase of αIgG-AF647-signal normalized to TagGFP gMFI upon Bafilomycin treatment. **i, j** Analysis of CD28-CD80- (**i**) and CTLA4-CD80- (**j**) stability upon acidification. 58αβ A2 T cells expressing CD28 or CTLA4 were incubated on ice with titrated amounts of CD80-Fc and subsequently treated with PBS (pH 7.2) or citric buffer (pH 5.5). After washing bound CD80-Fc was detected using αIgG-AF647 antibody by flow cytometry. See also Figures S5–8. Statistics: 6 h: Pooled data from 4 independent experiments. Two-sided paired $t$-test ($n = 4$ biologically independent samples per group). **i**: Representative data from 1 of 2 independent experiments. Extra sum-of-squares F test. $p$-value: $3 \times 10^{-15}$ ($n = 3$ biologically independent samples per group). **j**: Representative data from 1 of 2 independent experiments. Extra sum-of-squares F test ($n = 3$ biologically independent samples per group). Source data are provided as a Source data file.

cell endosomes or a combination of both. To distinguish between these scenarios, we used three different assays: First, for CTLA4-mediated trogocytosis, we demonstrate that ligand IC domain experiences an acidic pH—consistent with an endosomal rather than a cytosolic location. Second, a bi-molecular fluorescence complementation assay detected only very low levels of membrane fusion upon trogocytosis. Third, release of the ligand IC domain upon proteolytic cleavage of trogocytosed CD80 at the cell surface argues against its IC domain being located in the cytosol of the recipient cell. Together these three, independent results concordantly argue that trogocytosis involves little fusion of donor- and recipient cell membranes. Rather, they are in favor of the majority of trogocytosed ligands locating to endosomal ILVs, which may be secreted upon recycling to the cell surface. Such secreted microvesicles could at least in part remain attached to and decorate the cell surface. Consistent with this, electron microscopy of NK cells revealed that trogocytosed molecules locate to membranous structures on their surface[39]. Furthermore, it is well-established that T cells release microvesicles[40], which stimulate or inhibit neighboring T cell cells[41–43]. While traditionally inward budding of the endosomal limiting membrane is regarded to underlie the generation of endosomal ILVs and exosomes[44], our work supports trogocytosis as an additional mechanism for this.

An aspect that markedly differed between CD28 and CTLA4 was the fate of trogocytosed ligands. CD28 directed trogocytosed ligands to the cell surface. In contrast, CTLA4 promoted the lysosomal degradation of acquired ligands. Despite this dichotomy, our work neither negates that CD28-mediated trogocytosis can lead to some ligand-degradation, nor that some molecules trogocytosed via CTLA4 may recycle to the cell surface. Also repeated cycles of endocytosis driven by *cis*- or *trans*-interactions appear conceivable. Depending on the phosphorylation status of its intracellular domain CTLA4 either constantly cycles between the plasma membrane and different endosomal compartments or remains at the cell surface[24]. While cycling CTLA4 will drag trogocytosed ligands toward lysosomal degradation, at the cell surface—due to its high affinity—it will functionally silence them by *cis*-interaction. In contrast, ligands returning to the cell surface upon CD28-mediated trogocytosis co-stimulate neighboring T cells. This dichotomy may have important implications for communication among T cells. Specifically, our work complements multiple previous publications showing that trogocytosis of cognate pMHCs and subsequent presentation of acquired antigens enables T cells to activate neighboring T cells[45–48]. As availability vs. non-availability of co-stimulatory ligands is a decisive factor for activation vs. silencing of naive T cells, respectively, trogocytosis via CD28 and CTLA4 can be expected to form a crucial regulatory lever for such T-cell:T-cell antigen presentation. Furthermore, our work provides a proof of concept demonstrating how antagonistic receptors competing for shared ligands employ trogocytosis to differentially regulate ligand-availability. This concept may extend well beyond CD28 and CTLA4,

because at least 10 competing antagonistic receptor pairs are known, some of which also mediate trogocytosis[49].

Regarding mechanisms controlling the fate of endocytosed receptors, it is known that interactions between their intracellular domains and endosomal sorting machineries direct receptors to different endo-/ lysosomes and the cell surface[50]. For CD28 and CTLA4 several such interactions are known to mediate their predominant cell surface and endosomal location, respectively[21–24]. To investigate whether the different intracellular domains of CD28 and CTLA4 sufficed to explain the fate-divergence of trogocytosed ligands, we swapped the receptor intracellular domains. These experiments revealed that receptor intracellular domains directed trogocytosed ligands to more or less acidic compartments, but the receptor extracellular domains affected the extent of ligand-degradation. Receptor extracellular domains mediate ligand-binding and the stability of this interaction determines the extent to which ligands follow the trafficking receptor. Dissociation of receptor and ligand has been found to promote recycling to the cell surface, whereas stable interaction leads to lysosomal degradation[33,35,36,51]. Our observation that the receptor extracellular domain affected if trogocytosed ligands were degraded indicated that the stability of the receptor-ligand interaction may differ for CD28 versus CTLA4—and that this may be an important parameter underlying the fate-divergence of trogocytosed ligands.

The stability of receptor-ligand binding is subject to environmental conditions. Upon endosomal acidification pH-dependent, reversible protonation of histidines can modulate protein conformation and electrostatic interactions important for receptor-ligand interactions and may confer a pH-sensor function to proteins[52]. Upon comparing the CD28 and CTLA4 extracellular domains, we noted a striking paucity of histidines in CTLA4 compared to CD28. This led us to hypothesize that different acid-stabilities of CD28 and CTLA4 may underlie the fate-divergence of ligands trogocytosed by either receptor. Consistently, we demonstrated that at endosomal pH-levels CD80 dissociated from CD28, but not from CTLA4. Furthermore, we observed that CD80 poorly bound to an endosomally-targeted CD28-variant, which was restored by inhibiting endosomal acidification. We corroborated these results by different structural analyses, which revealed that acidification-induced charge alterations of CD28 affected both the ligand-interaction and the homodimer region. Specifically, in MD simulations we observed a conformational distortion of the CD28-CD80-interaction upon acidification, which moved the ligand-interaction motif of CD28 out of its pocket in CD80. This was substantiated by umbrella sampling simulations, which showed that acidification decreased the energy required to dissociate the monomeric CD80-CD28 interaction, whereas it had an opposite effect on CTLA4. In additional MD simulations, we observed that acidification destabilized the CD28 homodimer configuration without promoting a specific conformation of both monomers. Mutating both conserved histidines in the homodimerization region to its charged structural

analog arginine reduced CD28 surface expression, most likely due to increased ER retention. As ER export requires specific protein conformations, this supports the notion that acidification-induced histidine protonation promotes conformational flexibility of the CD28 homodimer. Experimentally we cannot reliably determine the relative contribution of acidification-induced effects on the CD28 ligand-interaction and homodimer regions to the dissociation we observed in wet-lab experiments, but consider it likely that both contribute to this. Consistent with experimental results, analogous structural modeling analyses of CTLA4 yielded results antithetical to those obtained for CD28. Specifically, charge distribution and molecular dynamic simulations concordantly indicated that acidification neither impairs the ligand-interaction nor the homodimer regions of CTLA4, and umbrella sampling simulations predicted the affinity of CTLA4 for CD80 to even increase upon acidification. Taken together, our work demonstrates that acidification-induced conformational changes of CD28 compromise its binding to CD80, thereby leading to ligand-release within endosomes and subsequent return to the cell surface. Conversely, the acid-stability of the CTLA4 conformation is important for its ability to direct trogocytosed ligands for lysosomal degradation. A proposition arising from this notion is that increasing the acid-stability of CD28 should result in more trogocytosed ligand becoming degraded. In this regard, the CTLA4/CD28(IC) hybrid receptor can be viewed as an affinity-enhanced CD28 variant, and indeed targeted more ligand for degradation than native CD28. Conversely, reducing the acid-stability of CTLA4 to CD80 should result in less degradation of trogocytosed CD80; and this is exactly what we observed upon comparing the CD28/CTLA4(IC) hybrid receptor to native CTLA4.

The current study did not directly investigate whether differences in signaling between CD28 and CTLA4 might affect the fate of trogocytosed ligands. Given the known interactions of the CD28 and CTLA4 intracellular domains with different trafficking regulators, this appears likely. Several publications described the sustenance of receptor signaling upon trogocytosis of ligands[53–55]. However, as sustained signaling is expected to require continuous receptor-ligand engagement, it will depend on the stability of this interaction within endosomes. Based on our results showing marked differences in the acid-sensitivity of the interaction between CD28 and CTLA4 with CD80, we expect that only CTLA4-signaling is sustained in acidic endosomes. In contrast, at the cell surface trogocytosed CD80 can also induce CD28 signaling, as shown by its ability to co-stimulate neighboring T cells. Whether it also enables cell-autonomous CD28 signaling remains to be determined.

Upon infections in vivo we found trogocytosis of CD80 and CD86 to predominantly occur in lymphatic organs, suggesting that the fate-separating effects of CD28 and CTLA4 on trogocytosed ligands regulate T cell behavior at these sites. Upon recognizing antigens presented by professional APCs T cells form cell clusters, in which they interact with APCs, but also with each other. Previously we have shown that within such cell clusters T cell expression of CD80 and CD86 enables them to collectively regulate their behavior by mutual stimulation and inhibition via CD28 and CTLA4, respectively[5]. Whereas CD28 is already present on naive conventional T cells, CTLA4 only appears more than 24 h after activation and its expression scales positively with T cell density and abundance. Hence, early after activation clustering T cells stimulate each other, whereas later when they have begun to expand increased levels of CTLA4 outcompete CD28 and lead to reciprocal inhibition among T cells. This tug-of-war is determined by the relative expression levels of CD28 and CTLA4 and the amount of available CD80 and CD86. Our work establishes trogocytosis as an additional layer regulating this network, by showing that it shapes the availability of CD80 and CD86. Specifically, CD28-mediated trogocytosis allows that early after activation clustering T cells do not only have to rely on self-synthesized CD80 and CD86, but in addition re-use ligands they have acquired from APCs for mutual stimulation. As displaying trogocytosed molecules at the cell

surface is a very rapid process compared to protein synthesis, this should accelerate immune responses. Once T cells express CTLA4, it not only competes with CD28 for access to CD80 and CD86, but also actively takes these ligands out of the competition via trogocytosis-mediated targeting for degradation. This dual function makes CTLA4 a more effective competitor for CD28. Taken together, our results indicate that the effects of trogocytosis of CD80 and CD86 on self-regulation of T cell population dynamics are contextual. Specifically, it is stimulatory as long as T cell density is low and few T cells express CTLA4, whereas it becomes more and more inhibitory once T cell density increases, which promotes CTLA4 expression. While our work delineates the mechanisms underlying trogocytosis and their impact on the fate of acquired proteins, the link of this process to collective behavioral regulation of T cells is largely descriptive. This is because it is currently not feasible to selectively perturb trogocytosis to directly assess its impact on T cell behavior without disrupting other essential endosomal trafficking processes. Nevertheless, on a broader scale our work showcases that regulating the availability of cell surface molecules is not solely a cell-autonomous process. Rather, trogocytosis trims cellular autonomy in favor of collective cellular regulation of protein availability, thereby adding an additional layer of control to quorum-regulation among cells.

## Methods

Research presented in this publication complies with all relevant ethical regulations. Animal experiments were approved by local governmental authorities (Regierungspräsidium Freiburg) and performed in accordance with EU guidelines.

### Mice

P14 TCR-transgenic mice (TcrLCMV)318Sdz/JDvsJ) have been described[56] (kindly provided by Hanspeter Pircher, University of Freiburg, Germany). *Cd80*−/− *Cd86*−/− (B6.129S4-*Cd80*tm1Shr *Cd86*tm2Shr/J), and mTmG (B6.129(Cg)-*Gt(ROSA)26Sor*tm4(ACTB-tdTomato,-EGFP)Luo/J) mice were from Jackson Laboratories and C57BL/6 mice from Janvier Labs. Splenocytes from *Cd28*−/− and *Ctla4*−/− OT-I mice were kindly provided by Niklas Beyersdorf and Monika Brunner-Weinzierl, respectively. All mice were on a C57BL/6-background and were housed under SPF-conditions. Animals at ages 6–22 weeks and of both sexes were used, but were sex-matched within experiments.

### Plasmids

Retroviral plasmids encoding murine *Cd80*, *Cd80*mut, *Cd86*, *Cd86*mut, and *Ctla4* have been described[5]. *CD28* was cloned from murine splenocyte cDNA. Receptors or ligands were tagged with fluorescent proteins (TagGFP, mScarlet, TagRFP, superecliptic pHluorin[27,57,58]) via flexible Glycine-Serine linkers. Variants of CD80 and CD86 impairing receptor-interaction (CD80 Y201A[59], CD86 Q35A[60]) were generated by side-directed mutagenesis. DNA encoding split-GFP subunits were ordered (Integrated DNA Technologies). Transgenes encoding CD28/Ctla4(IC) and CTLA4/CD28(IC) hybrid receptors were generated by overlap PCR. For these constructs the IC domains of CD28 (residues 178–218) or CTLA4 (residues 168–223) were attached to CTLA4 residues 1–193 or CD28 residues 1–183, respectively. This approach preserves the first few charged amino acids flanking the transmembrane domain, thereby promoting proper membrane integration. A cell surface-targeted, truncated CTLA4 variant (residues 1–193) lacking the internalization signals provided by the IC domain was generated accordingly. For the CD80-Ig construct we inserted amino acids 98–329 of murine IgG2a (Uniprot: P01863) flanked by Glycine-Serine-linkers between amino acids 1–246 and 243–306 of CD80 (Uniprot: (Uniprot: Q00609). DNA fragments were cloned into retroviral pMx-vectors[61] using NEBuilder® HiFi DNA Assembly reagents (New England Biolabs) and sequence-verified after cloning.

## Cells and cell culture

Chinese hamster ovary cells (CHO) were from ATCC (Cat.-No. CCL-61) and Platinum-A/E cells from Cell Biolabs (Cat.-No. RV-101, RV-102). SV40-immortalized murine embryonic fibroblasts (MEF), and 58αβ T cells were kind gifts from Hartmut Hengel and Wolfgang Schamel (both University of Freiburg), respectively. Cells lines stably expressing transgenes were generated by retroviral transduction (outlined below). All cells were cultured in IMDM, 10% FCS (PAN Biotech), L-Glutamin, PenStrep, 50 μM ß-Mercaptoethanol (all from Thermo Fisher Scientific) in 96-well u-, v- or flat-bottom plates or flasks (Greiner) at 37 °C, 5% CO$_2$ in a humidified environment. For Platinum-A/E 10 μg/ml blasticidin and 1 μg/ml puromycin (both from Applichem) were added. Dendritic cells were generated by culturing murine bone marrow in GM-CSF (20 ng/ml, Peprotech) for 5 days and matured by LPS (50 ng/ml, Sigma-Aldrich) for 16 h. Mature DCs were loaded with 10$^{-6}$M of LCMV gp33-epitope (amino acid sequence: KAVYNFATC) (PolyPeptide) for 1 h at 37 °C. B and CD8$^+$ T cells were magnetically purified from spleens (Mouse CD8 T lymphocyte enrichment set – DM, BD Biosciences and MojoSort™ Mouse Pan B Cell Isolation Kit, BioLegend). CD8$^+$ T cells were activated by addition of 10$^{-7}$ M gp33 peptide. For experiments depicted in Fig. 2f, g, *Cd80*$^{-/-}$*Cd86*$^{-/-}$ P14 CD8$^+$ T cells were co-cultured with WT P14 CD8$^+$ T cells at a 5:1 ratio to ensure their proper activation. In these experiments density of T cells was modulated by culturing them either in flat-bottom plates, which allow for a more scattered distribution, or in u- or v-bottom plates, in which gravity enforces higher cell densities. Protease sensitivity of surface receptors and trogocytosed molecules was assessed by treating cells with 0.05% trypsin for 30 min at 37 °C.

## Co-culture trogocytosis assays

1 × 10$^4$ MEFs transduced with fluorescently-tagged CD80 or CD86 were plated into 96-well flat-bottom plates. The next day 1 × 10$^5$ naive or transduced CD8$^+$ T cells or B cells were added and incubated for 2 h at 37 °C (unless stated otherwise). Alternatively, 3 × 10$^4$ 58αβ T cells were added to a monolayer of 4 × 10$^4$ CHO cells plated the day before into 96-well flat-bottom plates and co-cultured for up to 6 h. For DC-T cell co-culture 4 × 10$^4$ peptide-pulsed DCs were co-cultured with 10$^5$ P14$^+$ CD8$^+$ T cells for 24 h. In some experiments 100 nM Bafilomycin A1 (Selleckchem) was added. CTLA4-Fc recombinant protein (BioLegend) was used at 10 μg/ml. CD28 (clone E18, produced in house[62]) and CTLA4 (clone UC10-4F10, BioXcell) antibody Fab-fragments were generated using the Pierce® Fab preparation kits (Thermo Scientific, Langenselbold, Germany) according to the manufacturer's instructions[63]. For this, antibodies were first concentrated using protein ultrafiltration concentrators (Sartorius Stedim Biotech, Goettingen, Germany), then digested using immobilized papain under continuous mixing for 5 h at 37 °C and finally purified by protein A binding. Quality of Fab fragments was routinely tested by gel electrophoresis. Fab fragments were added 30 min prior to co-culture at 10 μg/ml.

## Co-culture of B and T cells

*Cd80*$^{-/-}$*Cd86*$^{-/-}$ B cells were transduced with CD28 or CTLA4 and then co-cultured with MEFs expressing CD80-mScarlet for 2 h. Then cells were harvested and labeled using anti-mouse CD106-APC antibody (BioLegend) selectively staining MEF cells. B cells were purified by a BD FACSAriaIII cell sorter (BD Biosciences) based on size and lack of CD106-expression. 10$^4$ sorted mScarlet$^+$ B cells were loaded with 10$^{-7}$ M gp33 peptide and co-cultured with 9 × 10$^4$ *Cd80*$^{-/-}$*Cd86*$^{-/-}$ P14 CD8$^+$ T cells. Where indicated, T cells were co-stimulated with agonistic CD28-antibodies at 1 μg/ml (clone 37.51, BioLegend).

## Generation of CD28-deficient 58αβ T cells

Crispr/Cas9-targeting of the CD28 locus in 58αβ T cells was performed using Crispr-Nanoblades[64]. gRNAs targeting *Cd28* exons 2 and 4 (ACATGAACATGACTCCCCGGAGG) were designed using vbc-score[65]. A single cell-derived clone (named 58αβ A2) showing complete loss of CD28-expression by FACS was used for experiments.

(GCTTGTGGTAGATAGCAACGAGG) and 4 (ACATGAACATGACTCCCCGGAGG) were designed using vbc-score[65]. A single cell-derived clone (named 58αβ A2) showing complete loss of CD28-expression by FACS was used for experiments.

## Retroviral transduction

Recombinant, replication-deficient retroviruses were generated by plasmid transfection of Platinum-E/A packaging cells using Fugene 6 (Promega). Supernatant was harvested after 24 h and/or 48 h and directly used. Transductions were performed by spin-infection (90 min, 2000 rpm, 30 °C) in non-tissue-culture treated 24-wells plates, pre-coated with retronectin (20 μg/ml, TaKaRa) overnight and blocked with 2% BSA (Roche) for 30 min. Prior to transductions purified CD8$^+$ T cells were activated with 50 ng/ml PMA + 0.5 μg/ml Ionomycin (both Sigma-Aldrich) and purified B cells with 10 μg/ml LPS (Sigma-Aldrich). Transduced cell lines were flow cytometrically purified.

## Adoptive cell transfer & infections

1 × 10$^3$ WT and *Cd80*$^{-/-}$*Cd86*$^{-/-}$ P14 CD8$^+$ T cells were adoptively co-transferred into sex-matched C57BL/6 mice. One day later mice were infected with 200PFU LCMV strain WE intravenously. Blood lymphocytes were enriched by erythrocyte lysis (BD FACS™ Lysing Solution, BD Biosciences). Lungs were perfused with PBS and digested with 300 U/ml collagenase II (Thermo Fisher Scientific) and 10 U/ml DNAse I (Sigma-Aldrich) in medium supplemented with 25 mM magnesium chloride (Sigma-Aldrich) for 1 h at 37 °C. Spleens, lymph nodes and livers were dispersed using 70 μm cell strainers (Greiner). Lymphocytes from liver and lung suspensions were isolated by density gradient centrifugation (Lympholyte®-M, Cedarlane).

## Flow cytometry

The following dye-conjugated antibodies were used: CD8α PerCP-eFluor 710 (clone 53-6.7, Thermo Fisher Scientific Cat#46-0081-80, 1:333), CD8α FITC (clone 53-6.7, Thermo Fisher Scientific Cat#11-0081-85, 1:400), CD8α APC (clone: 53-6.7, Thermo Fisher Scientific Cat#17008182, 1:250), CD25 BV785 (clone: PC61, BioLegend Cat#102051, 1:200), CD28 PE-Cy7 (clone: E18, BioLegend Cat#122014, 1:100), PE-Cy7 Mouse IgG2b Isotype Control (clone: MPC-11, BioLegend Cat#400325, 1:100), CD44 BV785 (clone: IM7, BioLegend Cat#103059, 1:1000), CD45R/B220 PerCP/Cyanine5.5 (clone RA3-6B2, BioLegend Cat#103236, 1:100), CD80 PE/Cy7 (clone 16-10A1, BioLegend Cat#104734, 1:200), Armenian Hamster IgG Isotype Control PE-Cy7 (clone: HTK888, BioLegend Cat#400921, 1:200), CD80 Brilliant Violet 421™ (clone 16-10A1, BioLegend Cat#104726, 1:200), Armenian Hamster IgG Isotype Control Brilliant Violet 421™ (clone: HTK888, BioLegend Cat#400935, 1:200), CD86 APC (clone GL-1, BioLegend Cat#105012, 1:100), Rat IgG2a Isotype control APC (clone: RTK2758, BioLegend Cat#400511, 1:100), CD90.1 (Thy1.1) PE (clone: OX-7, BioLegend Cat#202524, 1:2000), CD90.1 (Thy-1.1) Alexa Fluor 700 (clone: OX-7, BioLegend Cat#202528, 1:1000), CD90.2 (Thy-1.2) FITC (clone: 30-H12, Thermo Fisher Scientific Cat#11090385, 1:200), CD90.2 PE (clone: 30-H12, BD Biosciences Cat#553014, 1:1000), CD106 APC (clone: 429, BioLegend Cat#105717, 1:100), CD152 (CTLA4) PE (clone UC10-4B9, BioLegend Cat#106306, 1:200), Armenian Hamster IgG Isotype Control PE (clone: HTK888, BioLegend Cat#400907, 1:200), TCR Vα2 PerCP/Cyanine5.5 (clone: B20.1, BioLegend Cat#127814, 1:100). CD28-Fc, CTLA4-Fc, CD80-Fc (all from BioLegend) recombinant proteins were used at 10 μg/ml unless stated otherwise. Fc-tagged proteins were detected by fluorescent secondary antibodies anti-Human IgG Alexa Fluor 647 (Fcγ fragment specific, Jackson ImmunoResearch Cat#109-605-098, 1:400), anti-Human IgG Fc APC (clone: M1310G05, BioLegend Cat#410712, 1:20). For low pH ligand-dissociation experiments, CD80-Fc labeled cells were exposed to pH5.5 buffer (0.133 M citric acid, 0.066 M Na$_2$HPO$_4$) for 5 min. For

intracellular staining, cells were fixed with 2% formaldehyde solution (MERCK) diluted in PBS and permeabilized using Permeabilization Buffer (Thermo Fisher Scientific). Total cell numbers were quantified using counting beads (BioLegend). CTLA4 was stained for 30 min at 37 °C, whereas all other stainings were done at 4 °C. Live/dead viability dyes were from Thermo Fisher Scientific and BioLegend. Cells were analyzed on a BD Fortessa flow cytometer (BD Bioscience) using FACSDiva software v9. Unless stated otherwise, FACS-analyses of primary cells were performed on gated CD8+ or B220+ live cells. 58αβ cells were gated based on CD8 and Thy1.1 expression. Quantification of pHluorin/TagRFP gMFI ratio was performed after background subtraction. pH-calibration of trogocytosed pHluorin-TagRFP reporter molecules was performed using the Intracellular pH Calibration Kit (Thermo Fisher Scientific). Flow cytometry data analysis: FlowJo (BD Bioscience). Doublets and dead cells were excluded. Gating strategies are shown in Supplementary Fig. 9.

## Confocal imaging of trogocytosis

$10^4$ gp33 peptide-loaded MEFs expressing CD80-GFP were co-cultured with $5 \times 10^4$ CD8+ mTmG P14 T cells for 48 h on 0.01% poly-L-ornithine (Sigma) coated 35 mm μ-dishes (Ibidi) in phenol-red free medium. Alternatively, $10^5$ CHO cells expressing CD80-TagRFP were co-cultured with CD28- or CTLA4-TagGFP expressing 58αβ A2 T cells for 2 h in serum-free medium on poly-L-ornithine coated 35mm μ-dishes. Cells were fixed using 2% formaldehyde (MERCK) and analyzed using a Cell Observer SD spinning disk confocal microscope (ZEISS). Image analysis was performed using IMARIS 8.3.1 (Bitplane).

For co-localization studies, co-cultured cells were transferred to coated multispot microscope slides (Hendley-Essex) in serum-free medium. After adherence at 37 °C for 15 min cells were fixed in 2% formaldehyde for 15 min at RT. Cells were labeled using 5 μg/ml wheat-germ-agglutinin (WGA) lectin CF405S (Biotium). Note that addition of WGA lectin prior to fixation stains cell surface and some endosomal membranes, whereas after fixation it only demarcates the cell membrane.

Alternatively, after fixation cells were permeabilized with PBS 0.1% Saponin (Carl Roth) for 10 min, blocked with PBS 1% BSA (Sigma-Aldrich) and incubated with anti-LBPA antibody (6C4, Sigma-Aldrich Cat#MABT837, 1:100) in PBS 1% BSA for 30 min. Then cells were stained with an anti-Mouse IgG (H + L) Alexa Fluor Plus 405 (Thermo Fischer Scientific Cat#A48257, 1:400) secondary antibody and mounted in Mowiol (Sigma-Aldrich). Images were obtained on a Leica SP8 confocal LSM using a 63x oil immersion objective.

Co-localization of CD80 and LBPA or WGA lectin was analyzed using Volocity software (Perkin Elmer, Version 6.1) after setting uniform thresholds across different conditions. Co-localization was quantified using Pearson's coefficient for >20 cells from 2 experiments.

## Bioinformatic analysis and structure-based modeling

Charge distributions of published protein sequences (CTLA4: P16410, CD28: P10747, https://www.uniprot.org/) were analyzed using VOLPES (http://volpes.univie.ac.at). Sequences were aligned by Clustal Omega (https://www.ebi.ac.uk/Tools/msa/clustalo/) and visualized by Jailview v2.11.1.1 (https://www.jalview.org/). Isoelectric points of protein extracellular regions (obtained from UNIPROT) were calculated by protparam (https://www.expasy.org/resources/protparam).

The extracellular domains of CTLA4 and CD28 were reconstructed from PDB 1I85 (according to UNIPROT sequence P16410) and PDB 1YJD (according to UNIPROT sequence P10747), respectively. Interchain disulfide bonds were introduced between CTLA4 C157 and CD28 C141 residues.

Complexes of CD28 or CTLA4 with CD80 were constructed with one monomer of every protein. For CD28 and CTLA4 we only included the sequence resolved in the crystal structure (residues 19–136 and 38–155, respectively). The template for the CD80-CTLA4 complex was PDB 1I8L. This complex was used to build the CD80-CD28 complex by homology modeling. Umbrella sampling simulations of receptor-ligand interactions were performed using the CD80 N-terminal domain (residues 35–140). For analyzing the orientation of receptors (CD28 vs. CTLA4) and ligand (CD80) at different pH-levels, CD80 coordinates were fixed and normal vectors defined based on a plane whose orientation was defined by the XYZ-coordinates of Cα-atoms of three residues of CD28 and CTLA4 that are located in homologous positions in the crystal structures of both proteins (Supplementary Fig. 8f). For CD28 those were L38 (A), I109 (B) and Y122 (C) and for CTLA4 F56 (A), L126 (B) and Y139 (C). The unitary normal vector to the plane formed by these points was calculated using the following equations:

$$\mathbf{u} = \begin{bmatrix} x_B - x_C \\ y_B - y_C \\ z_B - z_C \end{bmatrix} \quad (1)$$

$$\mathbf{v} = \begin{bmatrix} x_A - x_C \\ y_A - y_C \\ z_A - z_C \end{bmatrix} \quad (2)$$

$$\mathbf{M} = \begin{bmatrix} M_x \\ M_y \\ M_z \end{bmatrix} = \begin{bmatrix} u_y \times v_z - u_z \times u_y \\ u_x \times v_y - u_y \times u_x \\ u_z \times v_x - u_x \times u_z \end{bmatrix} \quad (3)$$

$$\mathbf{M}' = \frac{\mathbf{M}}{\sqrt{M_x^2 + M_y^2 + M_z^2}} \quad (4)$$

Molecular dynamic simulations were carried out with GROMACS version 2016.5[66] using the gromos53a6 forcefield[67]. After initial minimization and equilibration steps (position restrained and free), systems were simulated with a 2 fs time step at 310 K and 1 bar using the velocity rescaling thermostat and the semiisotropic Parrinello-Rahman barostat[68]. Every system was simulated and analyzed for 50 ns. Independent simulations performed with the gromos53a6 and Charmm27 force fields yielded comparable results.

Umbrella sampling simulations were performed using the pull code implemented in GROMACS with a force constant of 1000 kJ mol$^{-1}$ nm$^{-2}$ and pulling rate of 0.001 nm per ps. For complexes with CD80, reaction coordinates were the distance between the Cα center of mass of interacting domains. Simulations were replicated >5 times. Energy profiles were calculated by jointly analyzing the potential of mean force required to increase the distance between receptor and ligand with the weighted histogram (WHAM) method[69,70]. Errors of energy profiles were estimated using the default bootstrap analysis option of the gmx-wham tool included in GROMACS.

For determining electrostatic properties of CD28 or CTLA4 complexed with CD80 crystals or modeled complexes were processed with the PDB2PQR v3.1.0 software and then analyzed with the Adaptive Poisson-Boltzmann Solver (APBS, Version 1.4.1)[71] calculating volume potentials at pH 5 and pH 7. Increase in electrostatic potential (ΔE) occurring upon changing pH was calculated as a voxel-wise difference of APBS results (pH 5 minus pH 7).

3D illustrations of protein structures were generated by PyMOL Version 2.0[72].

## Statistics

Statistical analysis: Prism 8 and 9 (Graph Pad). Choice of statistical tests was based on normality testing and correction algorithms for multiple comparisons were applied. Default setting of statistical tests was two-sided. In all figures error bars depict mean ± standard deviation. Figure layout: Inkscape (https://inkscape.org/).

**Reporting summary**

Further information on research design is available in the Nature Research Reporting Summary linked to this article.

## Data availability

All data generated or analyzed during this study are included in this published article and its supplementary information files. The data generated in this study are provided in the Supplementary Information/Source data file. Crystal structures of CTLA4 and CD28 extracellular domains were obtained from PDB (PDB 1I85 and PDB 1YJD, respectively). Amino acid sequences of CD28 and CTLA4 were obtained from uniprot (P16410 and P10747, respectively). Source data are provided with this paper.

## Code availability

Code used to calculate and compare APBS at different pH conditions in this study have been deposited in the Zenodo database under accession code https://doi.org/10.5281/zenodo.7010173. Code used to calculate orientation between proteins in a complex used in this study have been deposited in the Zenodo database under accession code https://doi.org/10.5281/zenodo.7010179.

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

## Acknowledgements

We acknowledge Clara Kamüller and Caro Schuler for mouse husbandry, Jan Bodinek for cell sorting and Jürgen Brandel for support on video compilation. We are grateful to Peter Aichele and Stephan Ehl for discussions and support. Funding for this research was provided by the Argentinian National Research Council (CONICET) (to C.R.S.), FONCYT (to C.R.S.: grant no. PICT-2018-01107), the German Research Foundation (DFG) (to J.C.R.: grant no. RO 4120/2-1 and RO 4120/3-1) and SFB1160 (to J.C.R.).

## Author contributions

Conceptualization: S.Z. and J.C.R. Methodology: A.H., A.A., M.B.W., R.E., N.B., and T.L. Investigation: S.Z., M.P.S., F.S., J.B., A.Z., A.G., and T.L. Visualization: S.Z., M.P.S., F.S., A.G., T.L., and C.R.S. Funding acquisition: J.C.R. and C.R.S. Project administration: J.C.R. Supervision: J.C.R. Writing—original draft: S.Z. and J.C.R. Writing—review & editing: S.Z. and J.C.R.

## Funding

## Competing interests

The authors declare no competing interests.
