## [Peer Review File · Nature Communications]

Differential trafficking of ligands trogocytosed via CD28 versus CTLA4 promotes collective cellular control of co-stimulationREVIEWER COMMENTS

Reviewer #1 (Remarks to the Author):

This is a very interesting paper and it adds significantly to our understanding of trogocytosis and the impact of checkpoint regulators on this process. The key findings include the differences in the mechanism of trogocytosis mediated by CTLA-4 and CD28 and the fate of the trogocytosed molecules. These findings fit into the existing models of trogocytosis (1-3) and expand our understanding significantly. The approaches were novel and were able to generate data which strongly supported the conclusions. This is potentially high impact manuscript that also provides additional information on the functional differences between CD28 and CTLA-4.

There were a couple of issues that I think need to be addressed, however. With the exception of figure 1, the experiments were not done in an antigen-specific system. This is an important limitation and I feel it needs to be addressed. The physiological relevance of CD28 or CTLA-4 mediating trogocytosis in the absence of antigen-driven activation of the cells is unclear. The impact of T cell receptor-mediated signaling and its integration with CTLA-4 or CD28 signaling in the process of trogocytosis and subsequent internal trafficking of the trogocytosed CD80 and/or CD86 is ignored in the vast majority of the data presented. Ideally this would be addressed via additional experiments, but I think that including a discussion of this issue in the discussion section is sufficient.

Related to this, the impact of trogocytosis-mediated signaling is not addressed. These molecules have been shown to engage cognate receptors on the trogocytosis-positive cell and mediate intracellular signaling (4-6). How might this signaling event contribute/provide context to the antigen-independent trogocytosis events examined in this study?

A final issue relates to the results in figure 2J. In the text it is implied that the after trogocytosis, T cells are able to present antigen to naïve T cells. This is not necessary a new finding as it has been reported previously (7-9), but as I understand it from the figure legend and methods section, this specific question is not even addressed in 2J. In that data, B cells were cultured with CD80 expressing MEFs and the B cells trogocytosed molecules from the MEF. After being pulsed with antigenic peptide, these B cells could stimulate the naïve T cells. The potential of T-T antigen presentation was not directly tested and it is, therefore, a stretch to conclude that T cells can stimulate naïve T cells. This limitation needs to be made clearer in their conclusions.

1. Martinez-Martin, N., E. Fernandez-Arenas, S. Cemerski, P. Delgado, M. Turner, J. Heuser, D. J. Irvine, B. Huang, X. R. Bustelo, A. Shaw, and B. Alarcon. 2011. T cell receptor internalization from the immunological synapse is mediated by TC21 and RhoG GTPase-dependent phagocytosis. *Immunity* 35: 208-222.
2. Stinchcombe, J. C., G. Bossi, S. Booth, and G. M. Griffiths. 2001. The Immunological Synapse of CTL Contains a Secretory Domain and Membrane Bridges. *Immunity* 15: 751-761.
3. Dopfer, E. P., S. Minguet, and W. W. Schamel. 2011. A new vampire saga: the molecular mechanism of T cell trogocytosis. *Immunity* 35: 151-153.
4. Zhou, J., Y. Tagaya, R. Tolouei-Semnani, J. Schlom, and H. Sabzevari. 2005. Physiological relevance of antigen presentosome (APS), an acquired MHC/costimulatory

complex, in the sustained activation of CD4+ T cells in the absence of APCs. *Blood* 105: 3238-3246.

5. Osborne, D. G., and S. A. Wetzel. 2012. Trogocytosis results in sustained intracellular signaling in CD4(+) T cells. *J Immunol* 189: 4728-4739.

6. Reed, J., M. Reichelt, and S. A. Wetzel. 2021. Lymphocytes and Trogocytosis-Mediated Signaling. *Cells* 10.

7. Helft, J., A. Jacquet, N. T. Joncker, I. Grandjean, G. Dorothee, A. Kissenpfennig, B. Malissen, P. Matzinger, and O. Lantz. 2008. Antigen-specific T-T interactions regulate CD4 T-cell expansion. *Blood* 112: 1249-1258.

8. Tatari-Calderone, Z., R. T. Semnani, T. B. Nutman, J. Schlom, and H. Sabzevari. 2002. Acquisition of CD80 by human T cells at early stages of activation: functional involvement of CD80 acquisition in T cell to T cell interaction. *J Immunol* 169: 6162-6169.

9. Boccasavia, V. L., E. R. Bovolenta, A. Villanueva, A. Borroto, C. L. Oeste, H. M. van Santen, C. Prieto, D. Alonso-Lopez, M. D. Diaz-Munoz, F. D. Batista, and B. Alarcon. 2021. Antigen presentation between T cells drives Th17 polarization under conditions of limiting antigen. *Cell Rep* 34: 108861.

Reviewer #2 (Remarks to the Author):

This report shows that CD80 and CD86 transferred to activated CD8+ T cells by trogocytosis exert co-stimulation to other T cells and that the molecules trogocytosed by binding to CD28 or CTLA-4 exhibit different trafficking in CD8 T cells. The experiments on the latter part are well conducted with various assays, although it is not clear what molecules are involved in the differences in the trafficking. The former part on immunobiological significance of the trogocytosis does not fully support the authors' claims, and needs to be strengthened.

Major comments:

1. Given that a trogocytotic event occurs between APC and T cells in contact, a key issue is whether the CD80 and CD86 molecules trogocytosed to T cells indeed bear the co-stimulatory capacity to stimulate other T cells in physiological settings. The results do not provide clear evidence. For example, In Fig. 2J, according to the figure legend, they used CD28- or CTLA-4-transduced CD80/CD86-deficient B cells to stimulate CD80/CD86-deficient P14 CD8+ T cells in vitro. Why B cells, not T cells? No data is provided regarding how much these B cells expressed trogocytosed CD80 molecules compared with similarly treated T cells. They should show the costimulatory effects with T cells expressing trogocytosed CD80/CD86.

2. Do the trogocytosed CD80/CD86 on CD8+ T cells form the immunological synapse (i.e., IS formation between TCR/CD28 of other CD8+ T cells and MHC class I along with trogocytosed CD80/CD86) to stimulate other T cells? Alternatively, do they provide a co-stimulation independent of IS formation? If the trogocytosed CD80/CD86 is able to stimulate other T cells, are they structurally similar to the original CD80/CD86, for example, the formation of a stable CTLA-4/CD80 lattice structure in IS?

3. In relation to the above issue, are other molecules, such as MHC/peptide complex or MHC alone, trogocytosed to T cells upon contact with APCs, as shown by Akkaya et al. (*Nat Immunol*. 2019)? Is this transfer of other molecules dependent on specific ligand-receptor interactions, in this case CD80/CD86 and CD28/CTLA-4? Does trogocytosis occur on CD11c and other proteins as well as lipid molecules expressed by APCs, as shown by a

recent report by others (Tekguc et al., PNAS 2021). If it is the case, how do these molecules behave in terms of possible ILV formation or membrane fusion in Figure 4?

4. In Figure 1E, does the result mean that CD8+ T cells can acquire CD80/CD86 from APCs without antigen stimulation? Are the authors looking at activated CD8+ T cells (as gated on CD44+ cells as in Fig. 1A)? If so, do they (also gp33-stimulated CD8+ T cells) express CTLA-4 in addition to CD28?

In a similar vein, in Figure 1B, when do they assess the acquisition of CD80 by CD8+ T cells after infection? If this is activation-dependent, the kinetics of CD80 expression by transferred CD8+ T cells needs to be shown. In addition, it may help to show the acquisition by non-transgenic CD8+ T cells (and also CD4+ T cells) in the hosts to assess antigen-non-specific acquisition of the molecules.

5. Regarding the different functional outcomes between CD28- versus CTLA-4-mediated trogocytosis in Figure 2, they discussed two possible mechanisms on page 7; i.e., CD28's more efficient directing of trogocytosed CD80/CD86 to the cell surface and silencing of CD80 trogocytosed via CTLA-4 due to cis-interaction of CD80 and CTLA-4. It is well known, however, that unlike CD28, the majority of CTLA-4 molecules are present intracellularly; they are more rapidly endocytosed by the aid of AP1/2; and TCR stimulation enables their membrane retention. Are these properties of CTLA-4 contribute to the different outcomes including the degradation of trogocytosed CD80/CD86?

6. In general, can their findings with transgenic CD8+ T cells be extended to CD4+ T cells especially in physiological settings? It need to be addressed at least in vitro.

Reviewer #3 (Remarks to the Author):

This is an interesting work that elucidates differential responses of CD28 and CTA4 receptors to CD80 ligand binding during trogocytosis in T-cells. The differences are shown to be pH dependent using novel reporter systems and MD simulations among others. Given these findings improve our understanding of T-cells, this manuscript shows potential impact and is likely to be of interest to the wide readership of Nature Communications. Molecular modeling methods provide some atomistic insights but the method details, discussions and conclusions from these simulations requires clarifications and significant revisions before the manuscript is ready for publication:

1) Page 17: His  Arg at an acid sensitive site in CD28 identified through an electrostatic analysis in this work shows increased flexibility between CD28:CD80 in MD. Cells transduced with this mutation shows reduced CD28 expression levels. The authors use this finding to support their discovery of acid sensitive regions in CD28 and the role of these regions for CD28 function and folding. However, this argument appears weak at it stands. This acid sensitive region mutation analysis should be accompanied by results with another mutation elsewhere, in a non-acid sensitive region, that either does not affect CD28 cell expression or even more clear link to interaction between CD28 and CD80. This could either be a follow-up experiment that the authors will run themselves, or at least discuss literature examples of such non-function inhibiting/affecting mutations outside the acid sensitive regions identified here. It is also important that the residue numbers for this HisArg be clearly described in the main text instead of referencing the supplementary figures where the reader needs to dig through the figure to find this information.

2) Page 16: Lack of complete disassociation of CD28:CD80 in MD is used to conclude that acidification only decreases affinity but not lead to complete disassociation. This is again a weak argument. First, it is unclear how long the MD has been run for (if it has been defined, it is not immediately apparent in either the methods section or elsewhere). Second, association/disassociation are kinetic events that are tricky to be characterized/observed using unbiased MD simulations and therefore the conclusion is a stretch based on the MD data. This conclusion must therefore be revised, or follow-up data must be presented to support this.

3) As a follow up to point 2 above, and as a general observation on the modeling results and discussion in manuscript: In methods, MD simulations are generically described to be run for >40ns. This is very ambiguous. The total simulation time for each MD run should be clearly defined in the main text. Also, the proportion/time-range of the production run MD trajectories that have been included or used for analysis should be clearly described.

4) Umbrella sampling simulations have been used to delineate the difference in stability of interactions between CD28:CD80 and CTLA4:CD80 across two pH ranges.

a. Units for Y-axis in Fig 6 should be double-checked: energy units should be kcal/mol not force units of kcal/mol/nm.

b. Convergence criteria used for umbrella sampling should be clearly described.

c. Error bars for the PMFs should be shown in Fig 6F. This is significant because with the error bars defined, energy barriers between pH 5 & 7 for CTLA4:CD80 could be significantly altered. This would probably necessitate revising the conclusion around difference in stabilization of CD28:CD80 and CTLA4:CD80 in response to pH changes.

5) Tip of normal vectors are used to visualize the destabilization of CD80:receptor interface across the different pH. Normal vectors must be properly defined and described.

Replies to REVIEWER COMMENTS:

We thank the reviewers for their detailed and constructive comments and for acknowledging our work as interesting, important and novel. Below we respond to their comments and explain how we have revised the manuscript to address the points raised.

Reviewer #1 (Remarks to the Author):

This is a very interesting paper and it adds significantly to our understanding of trogocytosis and the impact of checkpoint regulators on this process. The key findings include the differences in the mechanism of trogocytosis mediated by CTLA-4 and CD28 and the fate of the trogocytosed molecules. These findings fit into the existing models of trogocytosis (1-3) and expand our understanding significantly. The approaches were novel and were able to generate data which strongly supported the conclusions. This is potentially high impact manuscript that also provides additional information on the functional differences between CD28 and CTLA-4.

There were a couple of issues that I think need to be addressed, however. With the exception of figure 1, the experiments were not done in an antigen-specific system. This is an important limitation and I feel it needs to be addressed. The physiological relevance of CD28 or CTLA-4 mediating trogocytosis in the absence of antigen-driven activation of the cells is unclear. The impact of T cell receptor-mediated signaling and its integration with CTLA-4 or CD28 signaling in the process of trogocytosis and subsequent internal trafficking of the trogocytosed CD80 and/or CD86 is ignored in the vast majority of the data presented. Ideally this would be addressed via additional experiments, but I think that including a discussion of this issue in the discussion section is sufficient.

We agree with the implications raised by the reviewer's comment that T cells frequently do not only interact with a single type of ligand for the various receptors they express, and that interactions mediated by the T cell receptor (TCR) play a major role in regulating T cell behavior. That is why we performed the experiments depicted in Figure 1 in the presence of antigen recognized by the TCR and demonstrated that this augments trogocytosis of CD80 and CD86 (Figure 1E). For this increase in trogocytosis, we see at least three different, but non-mutually exclusive mechanisms:

First, as implied by the reviewer's comment, there are numerous publications describing how signaling via TCR, CD28 and CTLA4 impact on each other. Specifically, TCR-signaling induces a conformational change of CD28, which increases the avidity for its ligands (PMID: 24586641) and renders it more permissive for signaling (PMID: 29058713).

Second, it is well-established that TCR-mediated recognition of cognate pMHC presented by neighbouring cells stops the migratory behaviour of T cells and stabilizes the interaction with the antigen-presenting cell (PMID: 9108078). This prolongs contact times between cells, which can be expected to increase trogocytosis.

Third, the TCR itself mediates trogocytosis of pMHC-molecules (PMID: 10542149, 10553040, 10748232, 11238601). Importantly, several research groups have reported that TCR-mediated trogocytosis does not only transfer the pMHC-combination recognized by the TCR, but also extends to co-transfer of pMHCs that are not recognized by the TCR alongside the "cognate" pMHC (PMID: 16980510, 6804568, 15944248, 33730591). Based on these publications, we asked whether such TCR-mediated trogocytosis of "bystander" molecules also includes CD80 and CD86. To test this, we co-cultured naive P14 T cells for 2 hours with MEFs expressing CD80- or CD86-mScarlet in the presence or absence of the gp33 peptide recognized by the P14 TCR. To some samples, we had added soluble CTLA4-Ig, which interferes with the interaction between T cell expressed CD28 and MEF-expressed CD80 and CD86. As shown in Figure R1 below we observed that addition of CTLA4-Ig completely abrogated trogocytosis of CD80 or CD86 in the absence of a concomitant TCR-trigger. In contrast, in the presence of gp33 antigen, more T cells

trogocytosed CD80 or CD86, but addition of CTLA4-Ig was unable to completely block this, resulting in approximately 20% of cells still trogocytosing CD80 or CD86. These results indicate that concomitant TCR-triggering results in trogocytosis of CD80 and CD86 even if these ligands are not accessible to their receptor CD28, showing that TCR-mediated bystander trogocytosis also extends to CD80 and CD86. (Note that CTLA4 expression does not play a role here, as the 2 hour timeframe of this experiment does not suffice to induce CTLA4-expression in naive T cells). Conversely, our observation that a concomitant TCR-trigger was not required for CD28-mediated trogocytosis confirms previously published observations from Jonathan Sprent's group (PMID: 10748232 – Figure 2A).

Figure R1: MEF cells stably expressing CD80-mScarlet or CD86-mScarlet were antigen-loaded or not, and then co-cultured with P14 T cells with or without addition of CTLA4-Fc. After 2h T cells were harvested and analyzed by FACS for mScarlet expression. Note that concomitant TCR-triggering increased trogocytosis of CD80 and CD86, which could not be fully blocked by addition of CTLA4-Fc. In contrast, CTLA4-Fc completely abrogated trogocytosis of CD80 and CD86 in the absence of a concomitant TCR-trigger. Plot depicts pooled data from 2 independent experiments.

To test, whether bystander trogocytosis was confined to the TCR or whether it also applied to other receptors, we generated donor cells co-expressing an mScarlet-tagged CD86 mutant that is unable to bind CD28 and CTLA4 and a GFP-tagged CD80 (Figure R2). Upon co-culture with T cells we observed trogocytosis of both the wild-type and the mutant ligand. Taken together evidence from multiple experimental setups indicates that upon trogocytosis not only ligand recognized by receptors on the T cell surface are transferred, but that “neighboring”, non-binding cell surface molecules are dragged along. Such co-transfer of bystander molecules seems to be a general feature of trogocytosis and not confined to TCR-mediated trogocytosis. Conceptually, such transfer of bystander molecules suggests that trogocytosis includes transfer of membrane fragments surrounding recognized ligands (rather than solely the ligand itself).

Figure R2: Murine CD8⁺ T cells were co-cultured for 2 hours with MEFs expressing mScarlet-tagged native or mutated CD86, or co-expressing CD80-TagGFP and mutated CD86-mScarlet. Plot depicts mScarlet-fluorescence of CD8⁺ T cells. Note that T cells only acquire mutated CD86 if it is co-expressed together with CD80-GFP.

While all three TCR-driven mechanisms outlined in the above paragraphs (crosstalk of intracellular signaling, TCR-induced stop-signal and bystander trogocytosis) can be expected to modulate trogocytosis of CD80 and CD86, particularly TCR-driven bystander trogocytosis precludes a “clean” analysis of the effects mediated by CD28 and CTLA4. This is because when CD80 or CD86 are transferred via bystander trogocytosis, their fate is not determined by specific interactions with their receptors. As one aim of our work was to delineate the mechanisms underlying CD28- versus CTLA4-mediated trogocytosis, after having shown that TCR-triggering augments trogocytosis, in subsequent experiment we purposely chose a reductionist system without TCR-triggering and with cells expressing either CD28 or CTLA4. In order to clarify this important point, we have added the following sentences to

the manuscript: “This observation fits well to previous publications describing that TCR-triggering can induce trogocytosis of non-recognized molecules located in the vicinity of pMHCs recognized (PMID: 16980510, 6804568, 15944248, 27753080, 33730591). Upon such “bystander” trogocytosis, the fate of acquired molecules is not determined by specific interactions with their receptors. Based on this, we performed subsequent experiments without concomitant TCR-triggering. While this setup may be perceived as simplistic, by limiting confounding effects of bystander trogocytosis it allows for a clearer delineation of the mechanisms underlying trogocytosis of CD80/CD86.”

Admittedly the observation of CD28 or CTLA-4 mediated trogocytosis in the absence of antigen-driven T cell activation initially also came as a surprise to us. A careful look into existing literature revealed that for both CD28- and CTLA4-mediated trogocytosis analogous findings have been published by others (PMID: 10748232, 34301886). Hence, this seems to be a generalizable feature of CD28- and CTLA4-mediated trogocytosis that is not confined to our experimental system.

Regarding a potential physiological role for TCR-independent, CTLA4-mediated trogocytosis it may be noted that regulatory T cells have been found to suppress T cell responses even for antigens they do not recognize via their TCR (PMID: 10605010) and that CTLA4 expression is a crucial determinant of their function (PMID: 18845758). Hence, TCR-independent CTLA4 mediated trogocytosis directing CD80 for degradation would be a plausible mechanism to reconcile both observations, but this is still speculative.

For a potential physiological role of TCR-independent, CD28-mediated trogocytosis, it may be important to remind ourselves that while it is well-established that in the absence of a concomitant TCR-trigger CD28 signaling is not sufficient to drive activation of naive T cells, this does not allow to conclude that under these conditions CD28 is not able to exert any function at all. Nevertheless, we assume that in vivo CD28-mediated trogocytosis mainly occurs with concomitant TCR-triggering. We base this assumption on observations made by multiple research groups that in vivo conventional T cells do not make stable contacts with antigen-presenting cells unless they recognize the pMHCs presented by the APC. As direct cellular interaction is required for trogocytosis, we believe that the very short contacts T cells make upon “scanning” APCs for pMHCs may only allow for very little trogocytosis in the absence of TCR-mediated pMHC recognition. Once T cells recognize pMHCs presented by APCs they engage in longer lasting, more stable interactions with APCs, which facilitate trogocytosis. In contrast, in in vitro systems the architecture of cell culture wells promotes cellular interactions irrespective of pMHC-recognition.

All of this said, we fully agree with the reviewer that the physiological relevance of CD28 or CTLA-4 mediating trogocytosis in the absence of antigen-driven activation of the cells is unclear, which is why throughout our manuscript we did not emphasize any physiological role for this. Nevertheless, even if in vivo trogocytosis of CD80/CD86 mainly occurred in presence of a concomitant TCR-trigger, this does not refute the utility of TCR-independent reductionist systems to dissect and thereby elucidate fundamental mechanisms of trogocytosis, which are otherwise blurred in the complexity arising from multiple interactors acting simultaneously.

Related to this, the impact of trogocytosis-mediated signaling is not addressed. These molecules have been shown to engage cognate receptors on the trogocytosis-positive cell and mediate intracellular signaling (4-6). How might this signaling event contribute/provide context to the antigen-independent trogocytosis events examined in this study?

We thank the reviewer for raising this interesting point, which we indeed have not directly addressed in our manuscript. While for TCR-mediated trogocytosis of pMHC there is some evidence suggesting that it builds on TCR-signaling (PMID: 21820331, 11238601), for other receptors (namely KIR2DL1 and CTLA4) it has been shown that receptor mutants lacking the intracellular signaling domain can still mediate trogocytosis (PMID: 34301886, 17605758), arguing that receptor-signaling is not required for the actual molecule transfer in all receptors.

As pointed out by the reviewer several other labs have already shown that sustenance of receptor signaling upon interaction with endocytosed ligands can occur and modulate the behavior of the recipient cell. In our work we show that endosomal acidification promotes ligand-dissociation from CD28. Based on this it may be expected that CD28-mediated signaling is not sustained once trogocytosed ligands arrive in acidic endosomes. This could be different for CTLA4, whose ligand-binding we found to be retained under acidic conditions. Hence, it does not appear unlikely that CTLA4-mediated signaling continues even after endocytosis. Together these notions suggest that sustenance of signaling upon trogocytosis may be context dependent. While we agree that studying the signaling function of CTLA4 and CD28 in different endosomal compartments is an interesting topic to study, we believe that doing this in a thorough manner merits an entire project and publication, rather than merely an addition of a few experiments to our current publication. This said, in the revised version of our manuscript we have added these aspects to the discussion. Specifically, we included the following sentence(s): "Several publications have described the sustenance of receptor-signaling upon trogocytosis of ligands. As such sustained signaling is expected to require continuous receptor-ligand engagement, it will depend on the stability of this interaction within endosomes. While we did not directly investigate this, based on our results showing marked differences in the acid-sensitivity of the interaction between CD28 and CTLA4 with CD80, we expect that only CTLA4-signaling is sustained in acidic endosomes."

A final issue relates to the results in figure 2J. In the text it is implied that the after trogocytosis, T cells are able to present antigen to naïve T cells. This is not necessary a new finding as it has been reported previously (7-9), but as I understand it from the figure legend and methods section, this specific question is not even addressed in 2J. In that data, B cells were cultured with CD80 expressing MEFs and the B cells trogocytosed molecules from the MEF. After being pulsed with antigenic peptide, these B cells could stimulate the naïve T cells. The potential of T-T antigen presentation was not directly tested and it is, therefore, a stretch to conclude that T cells can stimulate naïve T cells. This limitation needs to be made clearer in their conclusions.

As pointed out by the reviewer TCR-mediated trogocytosis of pMHCs and subsequent presentation of acquired antigens resulting in activation of neighbouring T cells (T-T antigen presentation) has been shown in a number of previous publications (e.g. PMID: 25601867, 21242518, 15943619, 17442927). Therefore, in our original submission we did not feel that it was necessary to further substantiate this well-established point and, hence, did not address it in Figure 2J. However, as both reviewers 1 and 2 (comment 1) raised questions on the experimental setup used in this figure, we realized that we did not explain this point our reasoning for the chosen experimental setup sufficiently well, and we would like to apologize for this.

As suggested by the reviewers, initially we did perform analogous experiments to those depicted in Figure 2J using flow-cytometrically sorted antigen-stimulated Cd80^{-/-}Cd86^{-/-} T cells that had been co-cultured with MEFs expressing CD80 (or not) as stimulators for CFSE-labelled responder T cells. As shown in Figure R3 below, also in this setup responder T cells strongly proliferated upon interaction with stimulator T cells that had trogocytosed CD80. These results support our conclusion that T cells that have trogocytosed CD80 can utilize the acquired molecules to stimulate naïve T cells. We have now included this data into the manuscript.

Figure R3: T cell trogocytosis of CD80 enables co-stimulation of neighbouring cells. Antigen-stimulated Cd80^{-/-}Cd86^{-/-} T cells were co-cultured with CD80-mScarlet expressing MEFs, FACS-sorted for mScarlet-expression, gp33-antigen- loaded and co-cultured with CFSE-labeled Cd80^{-/-}Cd86^{-/-} P14 CD8⁺ T cells. Plots depict day 3 T cell CFSE dilution profiles. Plot depicts representative data of 2 independent experiments.

However, as outlined in the following paragraphs there are a number of crucial limitations inherent to this experimental setup, which is why we switched to the one depicted in Figure 2J and included this in the original submission.

Upon interaction of CD80 and CD86 with CD28 and CTLA4 the ligands are not only trogocytosed, but also induce signaling. In this regard it is well-established that TCR- and CD28-, but not CTLA4-signaling drives activated T cells to produce large amounts of IL-2 (by stabilizing cytokine mRNAs), which – in turn – affects T cell proliferation, apoptosis, differentiation and cytokine production. Hence, antigen-activated stimulator T cells having trogocytosed CD80 via CD28 will not only display CD80 to the responder T cells, but also secrete IL-2 (and TNF α and IFN γ). This will not be the case for stimulator T cells having trogocytosed CD80 via CTLA4. Hence, when using CD28- versus CTLA-4 expressing stimulator T cells the differential effects of CD28 and CTLA4 on the production of cytokines like IL-2 constitutes a confounder in the experimental system. Furthermore, the fact that IL-2 secretion occurs mainly within immunological synapses makes it extremely difficult to block its effect by adding antibodies.

A second confounding effect is related to TCR-triggering induced bystander trogocytosis of CD80 and CD86 (as outlined in our reply to the first comment of reviewer 1), which blurs the ability to clearly distinguish effects of CD28- versus CTLA4-mediated trogocytosis. As we and others (PMID: 10748232) have shown that TCR triggering was not required for CD28- and CTLA4-mediated trogocytosis, we set out to use an experimental system without TCR-triggering of the stimulator cells prior to FACS-sorting. In such a setup the use of T cells (that have trogocytosed CD80 or CD86) as stimulator cells is hampered by their poor survival in the absence of TCR-triggering. However, in accordance with previous publications (PMID: 21474713, 33039410), we found that engineering different cell types (B cells, T cell lymphoma cell lines, murine embryonic fibroblasts, chinese hamster ovary cells) to express CD28 or CTLA-4 was both necessary and sufficient to enable them to trogocytose CD80 and to direct them towards different fates. This argues that it is primarily the receptors CD28 and CTLA-4 and not some other “T-cell specific-factor” that is essential for this process.

Based on all these considerations, we choose to use primary B cells from Cd80^{-/-}Cd86^{-/-} mice, which lack expression of all 4 relevant interactors (CD80, CD86, CD28 and CTLA-4). Furthermore, it has been shown that B cells do not produce IL-2 unless stimulated by CD40 for long times, which will not be relevant in our systems as CD40-Ligand expression is largely confined to CD4⁺ T cells (PMID: 7539752). By transgenically expressing either CD28 or CTLA4 in primary B cells, we thus obtained a very “clean” system with comparable expression levels of CD28 and CTLA4 driven from the same promoter.

Since there are numerous publications showing that T cells synthesize and express CD80 and CD86 (PMID: 32049052, 7531145, 7679711), for obtaining a comparably clean system with primary T cells (which, in addition, express CD28 constitutively and CTLA4 and IL-2 upon activation) we would have had to generate quintuple knockout animals lacking expression of CD28, CTLA4, CD80, CD86 and IL-2. Otherwise, it would not have been possible to clearly distinguish the effects of endogenously and trogocytosed CD80 and of CD28 versus CTLA-4. Based on these considerations we felt that using primary B cells balanced the aspiration to demonstrate the differential effects of CD28- versus CTLA4-

mediated trogocytosis in primary lymphocytes, while at the same time not compromising the validity of the results.

In order to better explain the choice of experimental systems and the underlying line of reasoning we added the data depicted in Figure R3 into the manuscript and modified the paragraph corresponding to these experiments as follows: "We then tested the functionality of trogocytosed CD80 located on the recipient cell surface. For this, we co-cultured flow-cytometrically sorted activated P14 Cd80^{-/-}Cd86^{-/-} T cells that had trogocytosed CD80-mScarlet with naïve, CTV-labelled P14 Cd80^{-/-}Cd86^{-/-} T cells. As both cell populations are on a Cd80^{-/-}Cd86^{-/-} background any CD80 can solely derive from trogocytosed molecules. Since naïve T cells require concomitant TCR- and CD28-ligation, full T cell activation only occurs if trogocytosed CD80 is located at the cell surface and not blocked by cis-interactions. We observed that T cells that had trogocytosed CD80 efficiently activated naïve T cells (Suppl. Figure 2f). Potential confounders of this experimental setup are TCR-mediated bystander trogocytosis, which may affect the trafficking of trogocytosed CD80 and TCR-mediated production of IL-2 and other cytokines. Furthermore, the non-exclusive expression of CD28 and CTLA-4 by Cd80^{-/-}Cd86^{-/-} T cells does not allow to distinguish CD28- and CTLA4-mediated effects. To overcome these limitations, we modified the above experimental setup. For this we took into account that engineered expression of CD28 or CTLA-4 was both sufficient and necessary to enable different cell types to trogocytose CD80 (PMID: 21474713, 33039410). We transduced primary Cd80^{-/-}Cd86^{-/-} B cells, which lack endogenous expression of CD28 and CTLA4, with CD28 or CTLA4, let them trogocytose CD80-mScarlet, loaded them with antigen and co-cultured them with naïve, CTV-labelled P14 Cd80^{-/-}Cd86^{-/-} T cells. We observed that only cells trogocytosing CD80 via CD28, but not via CTLA4, efficiently activated naïve T cells (Figure 2J). Together these experiments confirm that CD28 and CTLA4 direct trogocytosed CD80 to different fates, that CD80 trogocytosed by CD28 is presented in a functional manner at the cell surface and that T cells can utilize trogocytosed CD80 to activate neighbouring naïve T cells."

To put these results into the context of T-cell:T-cell antigen presentation, we have added the following sentences to the discussion of the manuscript: "This dichotomy (of effects of CD28- vs. CTLA4-mediated trogocytosis) may have important implications for communication among T cells. Specifically, our work complements multiple previous publications showing that trogocytosis of cognate pMHCs and subsequent presentation of acquired antigens enables T cells to activate neighbouring T cells (PMID: 25601867, 21242518, 15943619, 17442927). As availability vs. non-availability of co-stimulatory ligand is a decisive factor for activation vs. silencing of naïve T cells, respectively, trogocytosis via CD28 and CTLA4 can be expected to form a crucial regulatory lever for such T-cell:T-cell antigen presentation."

1. Martinez-Martin, N., E. Fernandez-Arenas, S. Cemerski, P. Delgado, M. Turner, J. Heuser, D. J. Irvine, B. Huang, X. R. Bustelo, A. Shaw, and B. Alarcon. 2011. T cell receptor internalization from the immunological synapse is mediated by TC21 and RhoG GTPase-dependent phagocytosis. *Immunity* 35: 208-222.
2. Stinchcombe, J. C., G. Bossi, S. Booth, and G. M. Griffiths. 2001. The Immunological Synapse of CTL Contains a Secretory Domain and Membrane Bridges. *Immunity* 15: 751-761.
3. Dopfer, E. P., S. Minguet, and W. W. Schamel. 2011. A new vampire saga: the molecular mechanism of T cell trogocytosis. *Immunity* 35: 151-153.
4. Zhou, J., Y. Tagaya, R. Tolouei-Semnani, J. Schlom, and H. Sabzevari. 2005. Physiological relevance of antigen presentasome (APS), an acquired MHC/costimulatory complex, in the sustained activation of CD4+ T cells in the absence of APCs. *Blood* 105: 3238-3246.
5. Osborne, D. G., and S. A. Wetzel. 2012. Trogocytosis results in sustained intracellular signaling in CD4(+) T cells. *J Immunol* 189: 4728-4739.

6. Reed, J., M. Reichelt, and S. A. Wetzel. 2021. Lymphocytes and Trogocytosis-Mediated Signaling. *Cells* 10.
7. Helft, J., A. Jacquet, N. T. Joncker, I. Grandjean, G. Dorothee, A. Kissenpfennig, B. Malissen, P. Matzinger, and O. Lantz. 2008. Antigen-specific T-T interactions regulate CD4 T-cell expansion. *Blood* 112: 1249-1258.
8. Tatari-Calderone, Z., R. T. Semnani, T. B. Nutman, J. Schlom, and H. Sabzevari. 2002. Acquisition of CD80 by human T cells at early stages of activation: functional involvement of CD80 acquisition in T cell to T cell interaction. *J Immunol* 169: 6162-6169.
9. Boccasavia, V. L., E. R. Bovolenta, A. Villanueva, A. Borroto, C. L. Oeste, H. M. van Santen, C. Prieto, D. Alonso-Lopez, M. D. Diaz-Munoz, F. D. Batista, and B. Alarcon. 2021. Antigen presentation between T cells drives Th17 polarization under conditions of limiting antigen. *Cell Rep* 34: 108861.

Reviewer #2 (Remarks to the Author):

This report shows that CD80 and CD86 transferred to activated CD8+ T cells by trogocytosis exert co-stimulation to other T cells and that the molecules trogocytosed by binding to CD28 or CTLA-4 exhibit different trafficking in CD8 T cells. The experiments on the latter part are well conducted with various assays, although it is not clear what molecules are involved in the differences in the trafficking. The former part on immunobiological significance of the trogocytosis does not fully support the authors' claims, and needs to be strengthened.

Major comments:

1. Given that a trogocytotic event occurs between APC and T cells in contact, a key issue is whether the CD80 and CD86 molecules trogocytosed to T cells indeed bear the co-stimulatory capacity to stimulate other T cells in physiological settings. The results do not provide clear evidence. For example, In Fig. 2J, according to the figure legend, they used CD28- or CTLA-4-transduced CD80/CD86-deficient B cells to stimulate CD80/CD86-deficient P14 CD8+ T cells in vitro. Why B cells, not T cells? No data is provided regarding how much these B cells expressed trogocytosed CD80 molecules compared with similarly treated T cells. They should show the costimulatory effects with T cells expressing trogocytosed CD80/CD86.

As an analogous concern was raised by reviewer 1, we kindly refer the reader to our detailed reply to the last comment of reviewer 1 above.

2. Do the trogocytosed CD80/CD86 on CD8+ T cells form the immunological synapse (i.e., IS formation between TCR/CD28 of other CD8+ T cells and MHC class I along with trogocytosed CD80/CD86) to stimulate other T cells? Alternatively, do they provide a co-stimulation independent of IS formation? If the trogocytosed CD80/CD86 is able to stimulate other T cells, are they structurally similar to the original CD80/CD86, for example, the formation of a stable CTLA-4/CD80 lattice structure in IS?

The reviewer raises several interesting questions, which are currently unclear and technically very difficult to address. To answer these questions we consider it useful to first define the term "immunological synapse" (IS). A textbook definition of IS could be a non-permanent area of close-membrane contact between adjacent immune cells, which forms a prototypical ring-like structure with a peripheral adhesion domain and a centrally located area for antigen-presentation, co-stimulation and cytokine secretion, which

is associated with a reorganization of intracellular structures, like the microtubule-organizing center (MTOC). This rather holistic definition allows us to evaluate existing evidence for immunological synapses forming between T cells. Publications describing synapses between neighboring T cells showed close membrane contacts (detected by electron microscopy examination), reorientation of the MTOC and directed secretion of soluble factors between T cells (PMID: 18674934, 11728337, 30348790, 23475183, 19960307). However, whether or not these interactions contain other structural features classically associated with synapses between APCs and T cells (e.g. a prototypical ring-like structure with peripheral adhesion and central signaling and secretion domains (pSMAC, cSMAC), etc.) has to the best of our knowledge not been reported yet. Furthermore, also synapse detection based on imaging of CD80, TCR or CD28 have not yet been reported. Only one publication provides microscopical evidence of homogenous CD86 enrichment at cell-cell contacts between PMA plus ionomycin activated T cells, albeit at lower levels compared to immunological synapses forming between dendritic cells and T-cells (PMID: 18674934). Figure 1C of our manuscript shows that rather than being evenly distributed throughout the cell or on its surface, trogocytosed CD80 locates to distinct patches that are located at the contact interfaces of neighboring T cells, which would be consistent with it locating to immunological synapses forming between neighboring T cells. Beyond these co-stimulatory molecules, several publications have demonstrated that T cells can trogocytose entire pMHC-complexes from antigen-presenting cells and present them to neighboring T cells (PMID: 10542149, 10553040, 10748232, 11238601). In addition, many groups including ours have shown that T cells co-express the prototypical synapse-associated adhesion receptor-ligand pair LFA-1 and ICAM-1 (e.g. PMID: 23997225, 32049052). Hence, T cells possess the entire synapse-associated armamentarium of antigen-presentation (pMHC), co-stimulatory (CD80, CD86) as well as adhesion molecules (LFA-1, ICAM-1) – either derived from endogenous synthesis or acquired from APCs via trogocytosis. Another - albeit indirect - piece of evidence supporting the notion that T cells form synapses upon interaction with each other, is that fratricide among activated T cells has been found to depend on Perforin (PMID: 15307178), an effector molecule T cells secrete at immunological synapses. Taking all this information together, we consider it likely that immunological synapses containing pMHCs and co-stimulatory molecules can form between T cells, although we acknowledge that not all synapse-associated features mentioned in the definition above have been conclusively demonstrated. The ultrastructure and molecular distribution of immunological synapses (pSMAC, cSMAC, etc.) has been defined using TIRF-based super-resolution microscopy technologies, which – to the best of our knowledge - are applicable to lipid bilayer-based systems, but not to cellular interactions in 3D occurring in more physiological setups. This makes it technically very challenging to examine to which degree interactions formed between T cells display a similar architecture to those formed between T cells and APCs.

Regarding the question on structural similarity of trogocytosed to original CD80 and CD86, the resolution of microscopes, which are available to us, does not allow to clearly depict CTLA4/CD80 lattice structures. However, our analyses indicates that trogocytosed CD80/CD86 must at least retain so much structural similarity to the original molecule that these molecules retain the ability to bind to CTLA4-Ig (see Figure R4 below) and to co-stimulate neighboring cells (Figure 2J in the manuscript). While both results reinforce each other, we consider the ability to co-stimulate neighboring cells the stronger evidence and, hence, included this data in the manuscript.

3. In relation to the above issue, are other molecules, such as MHC/peptide complex or MHC alone, trogocytosed to T cells upon contact with APCs, as shown by Akkaya et al. (Nat Immunol. 2019)? Is this transfer of other molecules dependent on specific ligand-receptor interactions, in this case CD80/CD86 and CD28/CTLA-4? Does trogocytosis occur on CD11c and other proteins as well as lipid molecules expressed by APCs, as shown by a recent report by others (Tekguc et al., PNAS 2021). If it is the case, how do these molecules behave in terms of possible ILV formation or membrane fusion in Figure 4?

In accordance with several previous publications, we found that also pMHCs are trogocytosed by T cells, which is dependent on specific TCR-pMHC interactions (see Figure R5 below) (PMID: 10542149, 10553040, 10748232, 11238601), that simultaneous interaction of cells by different receptors (e.g. TCR and CD28) promotes the extend of molecule transfer (see Figure 1E of the manuscript) (PMID: 10748232) and also that bystander trogocytosis of molecules can occur without receptor engagement (see Figure R2 and our reply to the first comment of reviewer 1 above) (PMIDs: 16980510, 6804568, 15944248, 33730591). Furthermore, intercellular transfer of membrane lipids alongside trogocytosed proteins has also been shown in multiple publications (PMIDs: 11238601, 34301886, 19051238, 21604971). As all of these observations have already been published and in order to keep the manuscript concise and focused on those novel findings that are most relevant for understanding how the fates of trogocytosed molecules can be affected by the competing receptors CD28 and CTLA4, we kindly ask to not include these aspects into the current manuscript. Along this line, while we agree with the reviewer that elucidating how various trogocytosed cell membrane proteins (other than CD28 and CTLA4) and lipids behave in terms of membrane fusion and ILV formation (and why they do so) would be an interesting topic to study, we feel that this is beyond the scope of the current manuscript, as it would require a deep dive into membrane biophysics.

Figure R5: Trogocytosis of peptide-MHC-complexes by T cells was analyzed by co-culturing MEFs expressing a BFP-tagged H-2Db MHC class I molecule presenting gp33 peptide (gp33-BFP) with CD8+ T cells. Left plot: Time-course of gp33-MHC-BFP acquisition by CD8+ T cells expressing the gp33-specific P14 TCR. Right plot: 2h co-culture of P14 CD8+ cells with MEF gp33-BFP in absence or presences of anti-H-2Db blocking antibody and co-culture of OT-1 CD8+ cells with MEF gp33-BFP. All histograms depict BFP-signal of gated CD8+ T cells. Note that acquisition of pMHCs presenting gp33 required specific recognition by the TCR as it was abrogated by an antibody against H-2D^b and absent in OT-1 TCR expressing T cells.

4. In Figure 1E, does the result mean that CD8+ T cells can acquire CD80/CD86 from APCs without antigen stimulation? Are the authors looking at activated CD8+ T cells (as gated on CD44+ cells as in Fig. 1A)? If so, do they (also gp33-stimulated CD8+ T cells) express CTLA-4 in addition to CD28?

Yes, the results depicted in Figure 1E were obtained in a 2 hour co-culture experiment using naive CD8+ T cells without antigen stimulation. In Figure 1E no pre-gating on CD44+ T cells was performed. Regarding the question on CTLA4-expression: in previous experiments (shown in PMID: 32049052) we have extensively analyzed the expression dynamics of CTLA4 on CD8+ T cells using many different activation conditions and did not observe CTLA4-expression prior to 24 hours after the onset of activation. Hence, we can safely assume that CD8+ T cells in this 2 hour experiment did not express CTLA4. Regarding a discussion on CD28-/CTLA4-mediated trogocytosis of CD80/CD86 in the absence of concomitant TCR-triggering, we kindly refer the reader to our reply to the first comment of reviewer 1 on page 1 of this document.

In a similar vein, in Figure 1B, when do they assess the acquisition of CD80 by CD8+ T cells after infection? If this is activation-dependent, the kinetics of CD80 expression by transferred CD8+ T cells needs to be shown.

Regarding the in vivo experiments, we apologize for not having included information on the time at which the data was obtained, which was day 12 post infection. We have now included this information in the revised legend of Figure 1B. The acquisition is indeed activation-dependent – as supposed by the reviewer - and in the revised manuscript we show the requested kinetics of CD80 (and CD86) expression on transferred CD8+ T cells in Supplementary Figure 1. This shows that early after infection (day 4) substantially more trogocytosed CD80 and CD86 can be found on the T cell surface as compared to later time points (as evidenced by CD80 and CD86 expression on almost all Cd80^{-/-}Cd86^{-/-} T cells on day 4) (Figure R6).

Figure R6: Expression of CD80 and CD86 on adoptively transferred WT and Cd80^{-/-}Cd86^{-/-} P14 T cells upon LCMV-infection. Plots depict proportion of CD80+ (left) or CD86+ (right) cells in spleen pooled from 1-2 experiments per point in time. Statistics: Wilcoxon test.

In addition, it may help to show the acquisition by non-transgenic CD8+ T cells (and also CD4+ T cells) in the hosts to assess antigen-non-specific acquisition of the molecules.

Regarding this suggestion to show acquisition of CD80/CD86 by non-transgenic CD8+ T cells (and also CD4+ T cells) in the LCMV-infected mice to assess antigen-non-specific acquisition, there are some important caveats to consider. In these infection model experiments it would be straightforward to gate on endogenous, non-transgenic CD4+ or CD8+ T cells of the host animals and analyse their expression of CD80 and CD86. However, for these cells this does not allow to distinguish whether CD80 or CD86 expression was due to trogocytosis or due to endogenous synthesis of both molecules by the T cells themselves. In this regard it is worth to note that there are numerous publications by different laboratories

including ours, demonstrating that T cells themselves synthesize considerable amounts of CD80 and CD86 (PMID: 32049052, 7531145, 7679711, 9558065). Consistent with this, also the kinetic data shown in Figure R6 always shows higher expression levels of CD80 and CD86 on P14 compared to P14 Cd80^{-/-} Cd86^{-/-} T cells. This endogenous synthesis of CD80 and CD86 by T cells constitutes an important confounder and makes a reliable assessment of the acquisition of CD80 and CD86 by non-transgenic host T cells in this infection model impossible. However, in vivo data on trogocytosis by non-TCR-transgenic T cells is provided in PMID 10748232, in which the authors transferred polyclonal rat T cells into SCID mice and observed that both CD4⁺ and CD8⁺ T cells acquired murine CD80 and CD86 molecules. These results demonstrate that acquisition of CD80 and CD86 by T cells in vivo is not restricted to TCR-transgenic T cells.

Despite the use of non-transgenic, polyclonal T cells in the published experiments, the trogocytosis of CD80 and CD86 cannot unambiguously be regarded as antigen-non-specific. This is because one cannot exclude that those polyclonal T cells that did trogocytose did not concomitantly recognize pMHCs (e.g. alloantigens in the published rat into mouse transfer experiments). However, based on the fact that in vivo T cells do not make stable contacts with antigen-presenting cells unless they recognize the pMHCs the APCs present, we consider it likely that there is only little trogocytosis of CD80 or CD86 in the absence of concomitant TCR-triggering. This is most likely different in vitro where the architecture of cell culture wells enforces cellular interactions. Please also see our reply to the first comment of reviewer 1 (page 1 of this document) for a more detailed discussion of this topic.

5. Regarding the different functional outcomes between CD28- versus CTLA-4-mediated trogocytosis in Figure 2, they discussed two possible mechanisms on page 7; i.e., CD28's more efficient directing of trogocytosed CD80/CD86 to the cell surface and silencing of CD80 trogocytosed via CTLA-4 due to cis-interaction of CD80 and CTLA-4. It is well known, however, that unlike CD28, the majority of CTLA-4 molecules are present intracellularly; they are more rapidly endocytosed by the aid of AP1/2; and TCR stimulation enables their membrane retention. Are these properties of CTLA-4 contribute to the different outcomes including the degradation of trogocytosed CD80/CD86?

Yes, they absolutely do!

When comparing conventional antibody-based staining for flow cytometry on non-permeabilized and permeabilized T cells, CTLA4 is found to mainly reside in intracellular compartments with only a small fraction being located at the cell surface. However, such static "snapshot" assessments do not reveal an important part of CTLA4 biology, namely that it rapidly cycles between endosomal compartments and the cell surface. This can, for example, be detected by incubating T cells with fluorescent Anti-CTLA4 antibodies for 30 minutes at 37 degrees Celsius. In this type of "cycling stain" the antibody binds to CTLA4 when it comes to the cell surface and follows it back into intracellular compartments upon endocytosis. Using this kind of assay, it was found that even though at a given point in time only a small fraction of CTLA4 is located at the cell surface, this small fraction is constantly exchanged by CTLA4 molecules moving to and from the cell surface. This behavior results in a very large fraction of all CTLA4 molecules coming to the cell surface over relatively short time intervals. Upon binding to its ligands CD80 and CD86 CTLA4 can retain this cycling behavior, thereby directing trogocytosed ligands towards lysosomal degradation. Upon TCR-stimulation the CTLA4 intracellular domain is phosphorylated, which promotes longer residency times at the cell surface (PMID: 25538704). While this will compromise the ability of CTLA4 to promote the degradation of trogocytosed ligands, they are nevertheless functionally silenced because the higher affinity of CTLA4 effectively inhibits their interaction with CD28 present at the cell surface. Hence, for CTLA4 we found two different properties silencing trogocytosed CD80/CD86: Without concomitant TCR-trigger CTLA4 endocytoses CD80 and CD86 and directs them for lysosomal degradation. Upon TCR-triggering those CTLA4 molecules that stay at the cell surface may not immediately direct bound CD80/CD86 for degradation, but nevertheless block their interaction with CD28. Once the CTLA4 intracellular domain becomes dephosphorylated (e.g. due to cessation of TCR-

signaling) it will then switch to endocytosing bound ligands. In contrast to this, for CD28 we only found one mechanism, and this one makes acquired molecules available for co-stimulation of neighboring cells. To make this point clearer, we added the following sentences to the manuscript: “Depending on the phosphorylation status of its intracellular domain CTLA4 either constantly cycles between the plasma membrane and different endosomal compartments or remains at the cell surface (PMID: 9175836). While cycling CTLA4 will drag trogocytosed ligands towards lysosomal degradation, at the cell surface - due to its high affinity - it will functionally silence them by cis-interaction.”

6. In general, can their findings with transgenic CD8+ T cells be extended to CD4+ T cells especially in physiological settings? It need to be addressed at least in vitro.

Yes, the findings described for CD8+ T cells in our manuscript can indeed be generalized to non-TCR-transgenic CD4+ T cells. In PMID 10748232 the authors provide compelling evidence that in vivo both polyclonal CD4+ and CD8+ T cells trogocytose and present CD80 and CD86. Furthermore, in PMID: 12444120 it was demonstrated that human CD4+ T cells trogocytose CD80 from autologous APCs in vitro with and without TCR-triggering. In accordance with this, we observed that also murine polyclonal CD4+ T cells are able to trogocytose and present CD80 (Figure R7). Together these results show that T cell mediated trogocytosis of CD80 and CD86 is not restricted to TCR-transgenic CD8+ T cells, but extends to polyclonal human, rat und murine CD4+ T cells. If despite the fact that our data shown in Figure R7 merely confirms previously published results the reviewer or editor would like us to include it into our manuscript, we would be happy to do so.

Figure R7: Trogocytosis of CD80 by CD4+ T cells. Polyclonal murine T cells were co-cultured or not (control) with MEF cells stably expressing CD80-mScarlet. After 2h T cells were harvested and analyzed by FACS for mScarlet expression.

Reviewer #3 (Remarks to the Author):

This is an interesting work that elucidates differential responses of CD28 and CTA4 receptors to CD80 ligand binding during trogocytosis in T-cells. The differences are shown to be pH dependent using novel reporter systems and MD simulations among others. Given these findings improve our understanding of T-cells, this manuscript shows potential impact and is likely to be of interest to the wide readership of Nature Communications. Molecular modeling methods provide some atomistic insights but the method details, discussions and conclusions from these simulations requires clarifications and significant revisions before the manuscript is ready for publication:

1) Page 17: His  Arg at an acid sensitive site in CD28 identified through an electrostatic analysis in this work shows increased flexibility between CD28:CD80 in MD. Cells transduced with this mutation shows reduced CD28 expression levels. The authors use this finding to support their discovery of acid sensitive regions in CD28 and the role of these regions for CD28 function and folding. However, this argument appears weak at it stands. This acid sensitive region mutation analysis should be accompanied by results

with another mutation elsewhere, in a non-acid sensitive region, that either does not affect CD28 cell expression or even more clear link to interaction between CD28 and CD80. This could either be a follow-up experiment that the authors will run themselves, or at least discuss literature examples of such non-function inhibiting/affecting mutations outside the acid sensitive regions identified here. It is also important that the residue numbers for this HisArg be clearly described in the main text instead of referencing the supplementary figures where the reader needs to dig through the figure to find this information.

From this remark it appears as if we may not have presented the data sufficiently clear so that the reviewer may have misunderstood it: In the MD simulation of the H135R H140R mutant we analyzed the orientation and conformation of two CD28 monomers forming a dimer and not the “flexibility between CD28:CD80” – since these positions are not expected to affect the binding site of CD28 for CD80, but the CD28 homodimer interface. By fixing positive charges at the positions of two conserved histidine residues, H135R and H140R mimic effects of acidification-induced histidine protonation at the CD28 homodimerization region. Nevertheless, acidification also does impact on the CD28-CD80 interaction through electrostatic interactions as observed in the APBS analysis (Figure 6A) and in the MD simulations (Figure 6C), which is further supported by results from two different types of wet-lab experiments (Figures 6I, J).

To corroborate our conclusion on the role of acid-sensitive regions for CD28 function and folding the reviewer asks for an analysis and/or discussion of the effect of other mutations on CD28 expression or ligand-interaction. Mutations fulfilling the characteristics requested by the reviewer (i.e. not affecting acid sensitive residues, having a clear link to ligand-interaction, not affecting CD28 expression) have been described in PMID: 28711152. In this publication it was shown that different mutations of phenylalanine 51 of CD28 were expressed at a level comparable to native CD28, but induced stronger T cell proliferation due to higher ligand-binding affinity. Furthermore, in PMID 8649453 the expression of several CD28 mutants was investigated. Although expression was only reported in a qualitative manner, the authors described good expression of mutants affecting residues 29-30, 54-57, 71-73, 99-104, 104-106, 108-110, which, however, markedly differed in ligand-binding ability. Together these results show that CD28 expression can be tolerant to mutations even if they alter conformation in a way that impacts on ligand-interaction. In contrast, this was found to be different for the CD28 homodimerization region. Specifically, two publications from different labs reported that several mutations affecting CD28 homodimer formation (I91R, I114R, C123S and Y145L/Y150L) were all associated with impaired CD28 surface expression (PMID: 32765524, 15696168). This leads to the conclusion that CD28 is sensitive to conformational changes in its homodimerization region, whereas mutation in other regions are tolerated much better.

When we observed that our structural modeling predicted acidification to affect both charge (APBS) and conformational flexibility (MD simulation) of the CD28 homodimerization domain, we set out to perform experiments to corroborate these results. However, in wet-lab experiments it is technically not feasible to restrict the effect of acidification to a particular part of a protein. To overcome this limitation, we instead choose to modify CD28 conformation in a way analogous to the predicted effects of acidification through mutating its amino acid sequence. While the charges of most amino acids do not change much across a pH-range of 5-7, this is not the case for histidine, which switches from a non-protonated, uncharged state at pH7 to a protonated, charged state at pH5. As the CD28 homodimerization region contains two evolutionarily highly-conserved histidine residues in close proximity to each other (H135, H140), it can be assumed that these are major drivers of any acidification-induced effects on this region. Based on these considerations we choose to mutate these two histidine residues. From a structural point of view arginine mimics a protonated, charged histidine, which is why we choose to mutate the conserved histidine residues to arginine. To test whether these mutations affect CD28 conformational flexibility in a manner analogous to acidification, we again run molecular dynamic simulations, which corroborated this (shown in Figure S7B). When we expressed the CD28 H135R H140R double mutant in CD28-deficient cells, we found cell surface expression of the mutant to be lower compared to native CD28 (Figure S7C). It is well known that the conformation of newly synthesized proteins has to pass certain checkpoints, before they

are released from the endoplasmic reticulum (PMID: 15078901). Experimentally one can thus utilize ER retention as an indicator of altered protein conformation – given that mRNA expression levels are comparable. Using the IRES-Thy1.1 reporter as reference for comparable transduction efficacies and mRNA-transcription, our results indicate that only a part of all CD28 H135R H140R proteins achieve a conformation that allows expression at the cell surface. Importantly, the CD28 H135R H140R protein was not misfolded per se, in which case there would have been no CD28 expression at the cell surface. Rather the reduced expression level at the cell surface fits well to the notion that mutating H135 and H140 to arginine residues increases the flexibility of CD28, thereby decreasing the probability of achieving a conformation that enables efficient ER export. As the mutations are designed to mimic the effect of histidine protonation, this suggests that also acidification promotes conformational flexibility in the CD28 homodimerization region.

To further substantiate these results, we have generated single and double histidine mutants of CD28 and tagged them with GFP at their intracellular domain. Using confocal microscopy, we observed that native CD28 and CD28 H140R largely co-localized with the cell surface (co-localization of CD28-GFP with red cell surface depicted in yellow), whereas a substantial fraction of H135R containing variants displayed a patchy, intracellular localization pattern mutant (Figure R8a) – consistent with increased ER retention. We used flow cytometry to quantify cell surface expression levels of the different CD28-variants and found that those of the H140R mutant were comparable to wild-type CD28, whereas those of H135R were reduced, but not as strongly as those of the H135R H140R double mutant (Figure R8b). Together these results demonstrate that CD28 conformation can tolerate the impact of H140R as long as H135 is unaltered, but mutations at both sites synergize in impairing CD28 surface expression. From this we may infer that upon acidification-induced protonation both H135 and H140 contribute to destabilizing the CD28 homodimer configuration.

Figure R8:

(a) Confocal images of $Cd28^{-/-}$ T cells transduced to express CD28 variants fused to GFP at their C-terminus. Cell surfaces were stained with WGA-lectin and are depicted in red. Overlapping red and green signals are depicted in yellow. Note that for variants containing the H135R mutation a larger fraction of the GFP-signal is located intracellularly. Scale bar 5 μ m.

(b) Analysis of $Cd28^{-/-}$ T cells transduced with different CD28-IRES-Thy1.1 constructs. Note that despite comparable Thy1.1 expression, cell surface expression levels of CD28 variants are reduced.

To better explain the line of reasoning and to clearly state amino acid numbers, we have modified the manuscript text as follows: “Since we also identified an acid-sensitive site at the CD28 homodimer region centering around two evolutionarily conserved histidines (Suppl. Figure 7), we next analyzed the impact of endosomal pH on this region. MD simulations of CD28 homodimers indicated that acidification increased the flexibility of both monomers without promoting a specific conformation, thereby destabilizing the homodimer configuration (Figure 6g, Suppl. Figure 8b). We set out to test these simulations in wet-lab experiments by modifying the CD28 homodimer conformation in a way that mirrors the effects of acidification. For this, we reasoned that the conserved H135 H140 residues located in the CD28 homodimerization region are the major drivers of acidification-induced effects on this region. Hence, we replaced them by arginine, which mimics protonated histidine in terms of charge and structure. To verify this approach, we first ran further MD simulations, which showed that H135R H140R mutations increased the flexibility of the CD28 homodimer in a way analogous to acidification (Suppl. Figure 8b). Upon transducing this double arginine mutant in cells, we detected reduced expression levels at the cell surface compared to native CD28 (Suppl. Figure 8c). This was not due to less efficient transduction or lower mRNA expression as expression levels of the IRES-controlled Thy1.1 reporter were comparable between native and mutant CD28. Furthermore, when we expressed both variants with a GFP-variant fused to their intracellular domains, we found that expression of CD28 H135R H140R led to a more patchy, intracellular expression pattern compared to native CD28 (Suppl. Figure 8d). As newly synthesized proteins need to pass certain endoplasmic reticulum (ER) checkpoints, before they are released (37), this increased proportion of intracellular location can be taken as an indicator of altered protein conformation. Importantly, the CD28 H135R H140R protein was not misfolded per se, in which case there would have been no CD28 expression at the cell surface. Rather the reduced cell surface expression level fits well to the notion that mutating H135 and H140 to arginine residues increases the flexibility of CD28, thereby decreasing the probability of achieving a conformation which enables efficient ER export. As the mutations are designed to mimic the effect of histidine protonation, this supports the view that acidification promotes conformational flexibility in the CD28 homodimerization region. To further test the effects of acidification on CD28 conformation, we studied CD28 within endosomes. For this we exploited that in the CD28/CTLA4(IC) hybrid receptor the CTLA4-derived IC domain targets the CD28 extracellular and transmembrane domains to acidic endosomes. We expressed CD28/CTLA4(IC)-GFP or CTLA4-GFP fusion proteins and compared the staining intensity with CD80-Fc with or without blockade of endosomal acidification. For each receptor we normalized the CD80-Fc-derived signal to GFP expression to account for differences in protein expression levels (Figure 6h, Suppl. Figure 8e). For CTLA4, addition of Bafilomycin did not change the intensity of CD80-Fc-staining (reflected by a ratio of 1), indicating that its conformation is not altered by endosomal acidification. In contrast, Bafilomycin markedly increased CD80-Fc-staining of the CD28/CTLA4(IC) hybrid receptor, supporting the notion that inhibition of acidification reverses conformational changes CD28 undergoes in endosomes. This indicates that endosomal acidification destabilizes the conformation of CD28, but not of CTLA4, and that this impairs ligand-binding.”

2) Page 16: Lack of complete disassociation of CD28:CD80 in MD is used to conclude that acidification only decreases affinity but not lead to complete disassociation. This is again a weak argument. First, it is unclear how long the MD has been run for (if it has been defined, it is not immediately apparent in either the methods section or elsewhere). Second, association/disassociation are kinetic events that are tricky to be characterized/observed using unbiased MD simulations and therefore the conclusion is a stretch based on the MD data. This conclusion must therefore be revised, or follow-up data must be presented to support this.

MD simulations have been run and analyzed for 50 ns and we have now clearly stated this in the Materials & Methods section of the manuscript.

We fully agree with the reviewer that “association/disassociation are kinetic events that are tricky to be characterized/observed using unbiased MD simulations”. This is why in our original submission, we phrased our conclusion as a possibility (“while acidification-induced charge alterations at the receptor-ligand interface could decrease the affinity of monomeric CD28 to CD80, alone they may be insufficient to lead to complete dissociation.”) and not as a matter of fact. This said, to reduce the risk of misunderstanding we decided to remove this sentence.

Based on a number of studies that have investigated the relation between equilibrium association constant and binding strength, we consider it reasonable to propose that any perturbation on the interaction that decreases the affinity of the complex increases the probability of complete dissociation by pulling (PMID: 8298043, 25132724, 11371470, 8913624). The results of the structural modeling are corroborated by wet-lab experiments, in which we observed a dissociation of CD28 from CD80 upon acidification (Figure 6i). The MD simulations point towards potential mechanistic explanations for this observation by predicting that acidification-induced changes in histidine charge distort the monomeric interaction between CD80 and CD28 and increase the flexibility of the CD28 homodimer region. While both mechanisms could decrease the stability of the CD80:CD28 interaction in acidic environments, experimentally we cannot reliably determine if one of the two is a stronger driver of the dissociation we observed in wet-lab experiments, but consider it likely that both contribute to this. We have included these considerations in the discussion section of the manuscript.

3) As a follow up to point 2 above, and as a general observation on the modeling results and discussion in manuscript: In methods, MD simulations are generically described to be run for >40ns. This is very ambiguous. The total simulation time for each MD run should be clearly defined in the main text. Also, the proportion/time-range of the production run MD trajectories that have been included or used for analysis should be clearly described.

The simulations of all four complexes (CD28 or CTLA4 bound to the N-terminal domain of CD80, at low and neutral pH conditions) have completed 50 nanoseconds. We have now included this information in the manuscript by adding the following statement: “Every system was simulated and analyzed for 50 ns.”

4) Umbrella sampling simulations have been used to delineate the difference in stability of interactions between CD28:CD80 and CTLA4:CD80 across two pH ranges.

a. Units for Y-axis in Fig 6 should be double-checked: energy units should be kcal/mol not force units of kcal/mol/nm.

As correctly pointed out by the reviewer, the axis was indeed mislabeled. We have corrected this in the revised version of the manuscript and wish to apologize for this error.

b. Convergence criteria used for umbrella sampling should be clearly described.

Periodic boxes of the umbrella sampling simulations were dimensioned to ensure that the proteins can separate at a distance greater to the largest cutoff interaction without interacting with periodic images. With this criterium, once the proteins are out of their interaction range, pull force should tend to zero. This is what we observed in all cases. As an example, in Figure R9 we show the pull force profiles for 4 independent simulations of the CTLA4-CD80 complex. In addition, with the setting we used, the duration of simulations was more than twice the time necessary for the dissociation.

Figure R9: Pull force profiles of 4 independent umbrella sampling simulations of the CTLA4-CD80 complex

c. Error bars for the PMFs should be shown in Fig 6F. This is significant because with the error bars defined, energy barriers between pH 5 & 7 for CTLA4:CD80 could be significantly altered. This would probably necessitate revising the conclusion around difference in stabilization of CD28:CD80 and CTLA4:CD80 in response to pH changes.

As suggested by the reviewer we have revised the figure to include the error estimation of energy profiles obtained with the umbrella sampling simulations (displayed as shaded areas in Figure R10 below). This estimation was performed with a bootstrap analysis as described by Hub. et al {doi.org/10.1021/ct100494z}. In addition to the results shown in the manuscript using the GROMOS force field, we have repeated the umbrella sampling analysis with charmm27 and charmm36 force fields. In all cases we observed a significant decrease in the affinity of the CD28-CD80 complex in acidic conditions - supporting our conclusions.

Figure R10: Umbrella sampling simulations of monomeric receptor-ligand interactions.

5) Tip of normal vectors are used to visualize the destabilization of CD80:receptor interface across the different pH. Normal vectors must be properly defined and described.

In the revised version of the manuscript we have now included an illustration of the normal vector used to analyze the orientation of CD28 and CTLA4 receptors interacting with CD80 at different pH-levels as well as a description of its definition: For analyzing the orientation of receptors (CD28 vs. CTLA4) and ligand (CD80) at different pH-levels, CD80 coordinates were fixed and normal vectors defined based on a plane whose orientation was defined by the XYZ-coordinates of C-alpha-atoms of three residues of CD28 and CTLA4 that are located in homologous positions in the crystal structures of both proteins (Figure R11A below). For CD28 those were L38 (A), I109 (B) and Y122 (C) and for CTLA4 F56 (A), L126 (B) and Y139 (C). The unitary normal vector to the plane formed by these points was calculated using the equations depicted in Figure R11B below. Specifically, first we calculated two vectors "u" and "v" as the distance vectors between points C and B and between points C and A, respectively (Equations 1 and 2). These

vectors u and v are then fed into Equation 3 to calculate a cross product M . Finally, the tip of the normal vector (M') is the normalized M vector, according to Equation 4.

B

$$\mathbf{u} = \begin{bmatrix} x_B - x_C \\ y_B - y_C \\ z_B - z_C \end{bmatrix} \quad (1)$$

$$\mathbf{v} = \begin{bmatrix} x_A - x_C \\ y_A - y_C \\ z_A - z_C \end{bmatrix} \quad (2)$$

$$M = \begin{bmatrix} x \\ y \\ z \end{bmatrix} = \begin{bmatrix} \mathbf{u}_y \times \mathbf{v}_z - \mathbf{u}_z \times \mathbf{v}_y \\ \mathbf{u}_x \times \mathbf{v}_y - \mathbf{u}_y \times \mathbf{v}_x \\ \mathbf{u}_z \times \mathbf{v}_x - \mathbf{u}_x \times \mathbf{v}_z \end{bmatrix} \quad (3)$$

$$M' = \frac{M}{\sqrt{M_x^2 + M_y^2 + M_z^2}} \quad (4)$$

Figure R11: (A) Illustration of parameters and plane as well as (B) equations used to calculate normal vector of CD28 interacting with CD80.

REVIEWERS' COMMENTS

Reviewer #1 (Remarks to the Author):

This revised manuscript has addressed the issues I identified in the first review. The inclusion of additional data and additional text to clarify issues related to the rationale of focusing on CD28 and CTLA-4 in the absence of TCR stimulation have significantly improved the manuscript.

Several small issues remain, but none are sufficient to prevent publication in its current form. I recommend publication of this paper as it is.

The problems are relatively minor. My main issue was the possibility raised in the initial review that the trogocytosed CD80 could be engaging the CD28 and/or the CTLA-4 and that that signaling might be playing a role in the differential response to between the two receptors. In the rebuttal letter the authors made a reasonable claim that that was outside the scope of the current manuscript and added text to state that they thought only CTLA-4 would signal the cells based upon the acidic compartment data. While a reasonable conclusion for endosomal signaling, they also showed that the CD80 was re-expressed on the surface of the T cell after trogocytosis and was able to assist in activating other T cells, so it could also engage CD28 on the T cell and induce cell-autonomous signaling. While I would have liked to see them address this possibility, I don't think that it is necessary for them to assess it experimentally because it is outside the scope of the paper.

Reviewer #2 (Remarks to the Author):

They have responded to my comments properly.

Reviewer #3 (Remarks to the Author):

The authors have answered all my questions around their computational models and inferences from these models satisfactorily. I recommend publication.

Reply to reviewer`s comments:

Reviewer #1 (Remarks to the Author):

This revised manuscript has addressed the issues I identified in the first review. The inclusion of additional data and additional text to clarify issues related to the rationale of focusing on CD28 and CTLA-4 in the absence of TCR stimulation have significantly improved the manuscript.

Several small issues remain, but none are sufficient to prevent publication in its current form. I recommend publication of this paper as it is.

The problems are relatively minor. My main issue was the possibility raised in the initial review that the trogocytosed CD80 could be engaging the CD28 and/or the CTLA-4 and that that signaling might be playing a role in the differential response to between the two receptors. In the rebuttal letter the authors made a reasonable claim that that was outside the scope of the current manuscript and added text to state that they thought only CTLA-4 would signal the cells based upon the acidic compartment data. While a reasonable conclusion for endosomal signaling, they also showed that the CD80 was re-expressed on the surface of the T cell after trogocytosis and was able to assist in activating other T cells, so it could also engage CD28 on the T cell and induce cell-autonomous signaling. While I would have liked to see them address this possibility, I don't think that it is necessary for them to assess it experimentally because it is outside the scope of the paper.

We fully agree with the reviewer`s notion that ligation-induced signals are likely to affect the trafficking of trogocytosed ligands. As the intracellular signaling domains of CD28 and CTLA4 differ, one can expect that ligation of both receptors elicits different signals within cells. Such different signals may affect trafficking of both receptors as well as of ligands they have bound. The differences in ambient pH between the extracellular space and endosomes add another regulatory layer to this. Specifically, the acid-stability of the CD80-CTLA4 interaction indicates that CTLA4 signaling may be sustained in endosomes (allowing to continuously affect the trafficking of bound CD80). In contrast, the acid-sensitivity of the CD80-CD28 interaction suggests that CD28 signaling is not sustained in endosomes (and, hence, cannot continuously affect the trafficking of endocytosed CD80), but may be restricted to the cell surface. Data supporting the ability of trogocytosed ligands to elicit CD28 signaling at the cell surface is provided in experiments depicted in Figure 2J. While this experiment demonstrates this only for signaling between neighboring cells, we have little reason to believe that this could not also happen in a cell-autonomous way - although we have not formally shown this.

To emphasize these potential limitations of our work we have modified the manuscript text as follows (new parts in green):

The current study did not directly investigate whether differences in signaling between CD28 and CTLA4 might affect the fate of trogocytosed ligands. Given the known interactions of the CD28 and CTLA4 intracellular domains with different trafficking regulators, this appears likely. Several publications described the sustenance of receptor signaling upon trogocytosis of ligands (53-55). However, sustained signaling is expected to require continuous receptor-ligand engagement, which will depend on the stability of this interaction within endosomes. Based on our results showing marked differences in the acid-sensitivity of the interaction between CD28 and CTLA4 with CD80, we would expect that only CTLA4-signaling is sustained in acidic endosomes. In contrast, at the cell surface trogocytosed CD80 can also engage CD28, as shown by its ability to stimulate neighboring T cells. Whether this also enables cell-autonomous CD28 signaling remains to be determined.

Reviewer #2 (Remarks to the Author):

They have responded to my comments properly.

Reviewer #3 (Remarks to the Author):

The authors have answered all my questions around their computational models and inferences from these models satisfactorily. I recommend publication.